# Continuous Treatment of Refractory Wastewater from Research and Teaching Laboratories via Supercritical Water Oxidation–Experimental Results and Modeling

Mariana Bisinotto Pereira [1], Guilherme Botelho Meireles de Souza [1,2], Isabela Milhomem Dias [2], Julles Mitoura dos Santos-Júnior [3], Antônio Carlos Daltro de Freitas [3], Jose M. Abelleira-Pereira [4], Christian Gonçalves Alonso [2], Lucio Cardozo-Filho [1] and Reginaldo Guirardello [5,*]

1 Programa de Pós-Graduação em Engenharia Química, Universidade Estadual de Maringá (UEM), Avenida Colombo, 5790-Zona 7, Maringá 87020-900, PR, Brazil; bisinottoufg@gmail.com (M.B.P.); botelhg@gmail.com (G.B.M.d.S.); lcfilho@uem.br (L.C.-F.)
2 Programa de Pós-Graduação em Engenharia Química, Universidade Federal de Goiás (UFG), Avenida Esperança, s/n-Chácaras de Recreio Samambaia, Goiânia 74690-900, GO, Brazil; isabelamilhomem@ufg.br (I.M.D.); christian@ufg.br (C.G.A.)
3 Engineering Department, Exact Sciences and Technology Center, Federal University of Maranhão (UFMA), Av. dos Portugueses, 1966, Bacanga, São Luís 65080-805, MA, Brazil; jullesmitoura7@gmail.com (J.M.d.S.-J.); acd.freitas@ufma.br (A.C.D.d.F.)
4 Department of Chemical Engineering and Food Technology, Faculty of Sciences, International Excellence Agrifood Campus (CeiA3), University of Cádiz, 11510 Puerto Real, Cádiz, Spain; jose.abelleira@uca.es
5 School of Chemical Engineering, University of Campinas (UNICAMP), Av. Albert Einstein 500, Campinas 13083-852, SP, Brazil
* Correspondence: guira@feq.unicamp.br

**Abstract:** Teaching and research laboratories generate wastes of various compositions and volumes, ranging from diluted aqueous solutions to concentrated ones, which, due to milder self-regulation waste-management policies, are carelessly discarded, with little attention given to the consequences for the environment and human health. In this sense, the current study proposes the application of the supercritical water oxidation (SCWO) process for the treatment of complex refractory wastewater generated in research and teaching laboratories of universities. The SCWO, which uses water in conditions above its critical point (T > 647.1 K, $p$ > 22.1 MPa), is regarded as an environmentally neutral process, uniquely adequate for the degradation of highly toxic and bio-refractory organic compounds. Initially, the wastewater samples were characterized via headspace gas chromatography coupled with mass spectrometry. Then, using a continuous tubular reactor, the selected operational parameters were optimized by a Taguchi L9 experimental design, aiming to maximize the total organic carbon reduction. Under optimized conditions—that is, temperature of 823.15 K, feed flow rate of 10 mL min$^{-1}$, oxidizing ratio of 1.5 (50% excess over the oxygen stoichiometric ratio), and sample concentration of 30%—TOC, COD, and BOD reductions of 99.9%. 91.5% and 99.2% were achieved, respectively. During the treatment process, only $CO_2$, methane, and hydrogen were identified in the gaseous phase. Furthermore, the developed methodology was applied for the treatment of wastewater samples generated in another research laboratory and a TOC reduction of 99.5% was achieved, reinforcing the process's robustness. A thermodynamic analysis of SCWO treatment of laboratory wastewater under isothermal conditions was performed, using the Gibbs energy minimization methodology with the aid of the GAMS® 23.9.5. (General Algebraic Modeling System) software and the CONOPT 4 solver. Therefore, the results showed that SCWO could be efficiently applied for the treatment of wastewater generated by different teaching and research laboratories without the production of harmful gases and the addition of hazardous chemicals.

**Keywords:** supercritical water; refractory effluent; organic degradation; thermodynamic analysis

## 1. Introduction

Teaching and research laboratories conduct experiments aimed at educating individuals, conducting investigations, and providing services. For such experiments, several types and properties of chemical products are employed. In general, all laboratory activities generate waste, ranging from diluted aqueous solutions to concentrated and/or recalcitrant ones. Commonly, the effluents from teaching and research laboratories are generated by the mixture of small quantities of different wastes, resulting in a wide diversity of composition and volume produced [1].

In general, wastewater management policies for profitmaking entities are strict and demanding, while non-profit organizations, such as most universities, follow milder self-regulation, due to their reduced size [2]. In the recent past, wastewater generated in teaching and research laboratories was carelessly discarded, with little to no regard for the consequences for the environment and human health. At that point, these behaviors were justified by two reasons: (i) it was believed that dilution could minimize, or even neutralize, the harmful potential of the discarded waste and (ii) the amount of chemical products used, and their effects, were considered not significant. Currently, the awareness of the importance of efficient methods for the treatment of wastewater generated in teaching and research laboratories is growing rapidly. Many studies have highlighted the necessity of appropriate practices in wastewater management in laboratories, ensuring environmental safety and public health [3].

Liquid effluents are traditionally treated through two distinct methods: biological methods and physicochemical processes. Each treatment method presents its own set of advantages and limitations, and the choice of the most suitable method depends on numerous factors, including the effluent characteristics, operational restrictions, and costs. Due to the use of microorganisms, biological processes are not suitable for the treatment of toxic effluents or those contaminated with recalcitrant substances of low biodegradability [4]. For such situations, advanced physicochemical processes stand out.

Advanced processes can be defined as those capable of producing higher-quality effluents, typically achieved through the combination of a series of treatment techniques. In those cases, these processes assist in the additional removal of pollutants present in low concentrations, which are hardly removed through conventional treatments. Due to the increased demand for clean water and the emergent necessity to reuse this finite resource, some advanced treatment processes based on physical–chemical phenomena have been evaluated, such as adsorption [5], membrane filtration [6], and advanced oxidation processes (AOPs).

AOPs are promising techniques for wastewater treatment, since they do not present any remaining toxicity, have high applicability, and can be considered environmentally sustainable processes [7]. AOPs include highly reactive oxidizing agents that can oxidize different organic pollutants [8]. In theory, AOPs could completely mineralize organic compounds into carbon dioxide ($CO_2$) and water ($H_2O$), according to Equation (1) [9].

$$R - H + HO \bullet \rightarrow H_2O + R \bullet \qquad (1)$$

Among the several AOPs studied in the last few years, the supercritical water oxidation (SCWO) process stands out [10,11]. This technique operates at conditions above the critical point of water (T > 647.1 K, $p$ > 22.1 MPa) and it is regarded as an environmentally neutral process, since it does not require hazardous and pollutant chemical additives [12]. Water does not exhibit toxicity, flammability, or any adverse effects to human health, being considered an environmentally universal solvent. In the SCWO process, water acts both as a reactant and as the reaction medium [13,14]. Above its critical point, water is uniquely adequate for the degradation of highly toxic and biorefractory organic compounds present in both liquid effluents and solid waste [15]. In general, pollutants consisting only of the elements carbon, hydrogen and oxygen could be easily degraded into smaller and harmless molecular compounds [16].

According to Qian et al. [17], the supercritical water process can be classified into three methods, based on the oxidizing ratio. The oxidizing ratio is the relationship between the amount of oxidant added and the amount of oxidant theoretically required. When n = 0, the method is called supercritical water gasification. For 0 < n < 1, it is named supercritical water partial oxidation. For n $\geq$ 1, the method is referred to as supercritical water oxidation. The first and second methods are widely applied to $H_2$ production [18], while the latter is regarded as an environmentally friendly treatment for organic pollutants [19].

The degradation of different types of dyes and other organic substances (imidazoline, acetic acid, and dimethyl coco-benzyl ammonium-chloride) frequently found in textile industry wastewater was investigated by Sogur and Askgun [20]. Using a tubular reactor, the system reached a complete TOC removal (100%) at a temperature of 823.15 K in the presence of $H_2O_2$ (17.73 mmol $L^{-1}$) and with a residence time of 10 s.

In 2010, Youngprasert et al. [21] investigated the degradation of model laboratory wastewater containing acetonitrile via supercritical water oxidation using a compact-sized tubular reactor (internal volume of 4.71 mL). Manganese dioxide and $H_2O_2$ were used as the catalyst and oxidant, respectively. The complete oxidation of the acetonitrile was achieved at 673.15 K, 25 MPa, and feed flow rate of 2 mL $min^{-1}$. $N_2$, $CO_2$, and CO were observed as the main components of the gaseous phase. In 2016, Ferreira-Pinto et al. [22] reported experimental data on TOC reduction via SCWO from a model of dairy industry wastewater (lactose). Under constant pressure of 22.5 MPa, a feed flow rate of 5 mg $L^{-1}$, a temperature of 823.15 K, and the presence of hydrogen peroxide, a continuous tubular achieved a TOC reduction of 99%.

In 2017, Roshchin et al. [23] observed that the supercritical water degradation of organic compounds—specifically, well-known pesticides such as dichlorodiphenyltrichloroethane (DDT), hexachlorobenzene (HCB), and hexachlorocyclohexane (HCH)—is highly dependent on the presence of an oxidizing agent. The addition of air as an oxidizing agent enhanced the degree of decomposition of DDT, HCB, and HCH from 75%, 52%, and 40% (at 923.15 K) to 99.8%, 99.7%, and 99.9% (at 823.15 K), respectively. In 2019, Mylapilli and Reddy studied the supercritical water oxidation of pharmaceutical-industry wastewater containing analgesic, antibiotic, antipyretics, and antifungal properties, with an initial total organic carbon (TOC) concentration of 2017 mg $L^{-1}$. Under optimal conditions—that is, a temperature of 823.15 K, pressure of 23 MPa, residence time of 60 s, and the presence of $H_2O_2$—a reduction of 97.8% of the TOC was achieved [13].

In this sense, it remains clear that the recent research on the use of the supercritical gasification for treatment purposes has been focusing on wastewater generated by large scale industries [16]. On the other hand, when laboratory wastewater was investigated, only model solutions were assessed at a reduced scale and extremely low concentrations [21]. Therefore, the current study proposed the application of the SCWO for the continuous treatment of real organic refractory wastewater generated at the research/teaching laboratories of two Brazilian universities. Additionally, a thermodynamic simulation was conducted to determine the multiple component/phase system equilibrium.

## 2. Materials and Methods

### 2.1. Materials

Samples of refractory wastewater were collected from the Department of Chemical Engineering at the State University of Maringá (Brazil) and from the Department of Chemistry at the Federal University of Goiás (Brazil). The effluent was generated at a laboratory that specialized in high-performance liquid chromatography (HPLC) analyses. Initially, due to the limited information found on the labels of the wastewater bottles, as depicted in Figure 1, the effluent was characterized using headspace gas chromatography coupled with mass spectrometry (headspace GC–MS).

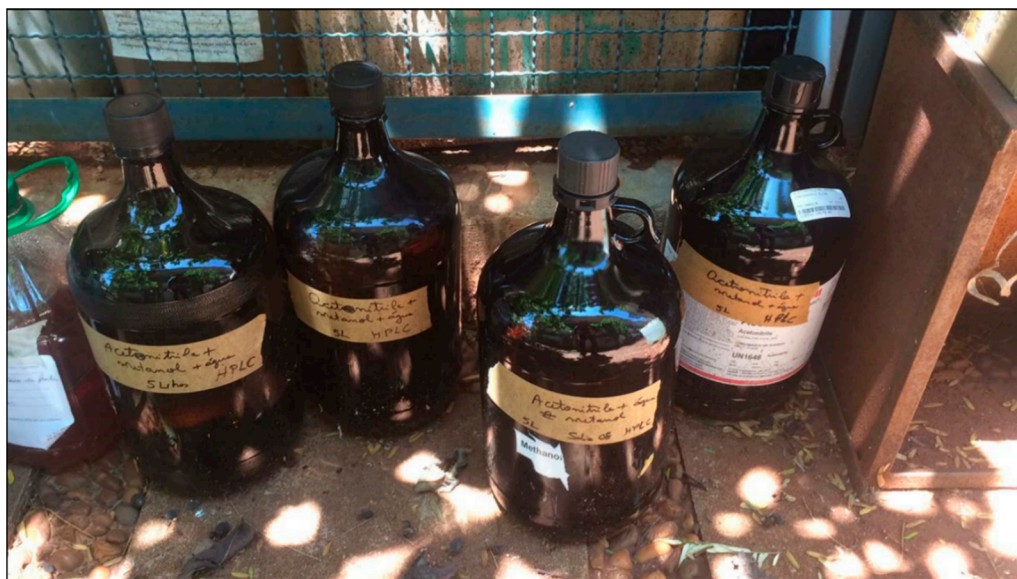

**Figure 1.** Samples of refractory wastewater at the Department of Chemical Engineering at the State University of Maringá (Brazil).

The detailed characterization of the refractory wastewater collected at the Department of Chemical Engineering at the State University of Maringá (Brazil) is summarized in Table 1.

**Table 1.** Main physical–chemical parameters of refractory wastewater samples.

| Parameters | Concentration (mg L$^{-1}$) |
|---|---|
| TOC | 47,980 |
| COD | 204,876 |
| BOD | 489,056 |

### 2.2. Experimental Procedure

The experimental apparatus was operated in a continuous mode. The setup was previously presented in detail by de Souza et al. [24]. In short, the system consists of a high-pressure pump, a preheater, a tubular reactor, a heat exchanger/condenser, a back-pressure regulator (BPR), and a phase separator. The feedstock solution was composed of a diluted sample of refractory wastewater in certain number of experiments, with $H_2O_2$ as an oxidizing agent.

Initially, the refractory solution was continuously fed by a high-pressure isocratic pump into the coiled preheater. Then, the solution was preheated to a temperature of 623.15 K. The preheating process was necessary to ensure that, upon entering the reactor, the solution rapidly reached the supercritical condition at the desired temperature. Subsequently, the solution was sent to the tubular reactor (Inconel 625 alloy) with an inner diameter of 13 mm, an outer diameter of 40 mm, and a length of 373 mm. Both the preheater and the reactor were heated by split furnaces that were internally filled with rock wool and equipped with two 1000 W infrared ceramic heaters. The preheater temperature and the tubular reactor temperature were monitored and controlled using J-type thermocouples. The system pressure was kept constant at 25 MPa by a manual BPR and monitored by a pressure gauge. After the supercritical water oxidation reaction, the treated solution was sent to a heat exchanger, cooled through a coiled-type condenser, and maintained at 278.15 K using a thermostatic bath. Finally, the liquid and gas phases were continuously collected at the end of the system, respectively, at the bottom and top of a phase separator. In general, after the desired operation conditions of feed flow rate, temperature, and pressure were reached, the system was operated over a period of 1 h for each experimental

run to allow for the accumulation of the liquid treated solution and analysis of the gaseous products. Then, at the end of the experiment, the tubular reactor was cooled to room temperature and depressurized. A graphic description of the research methodology is presented in Figure 2.

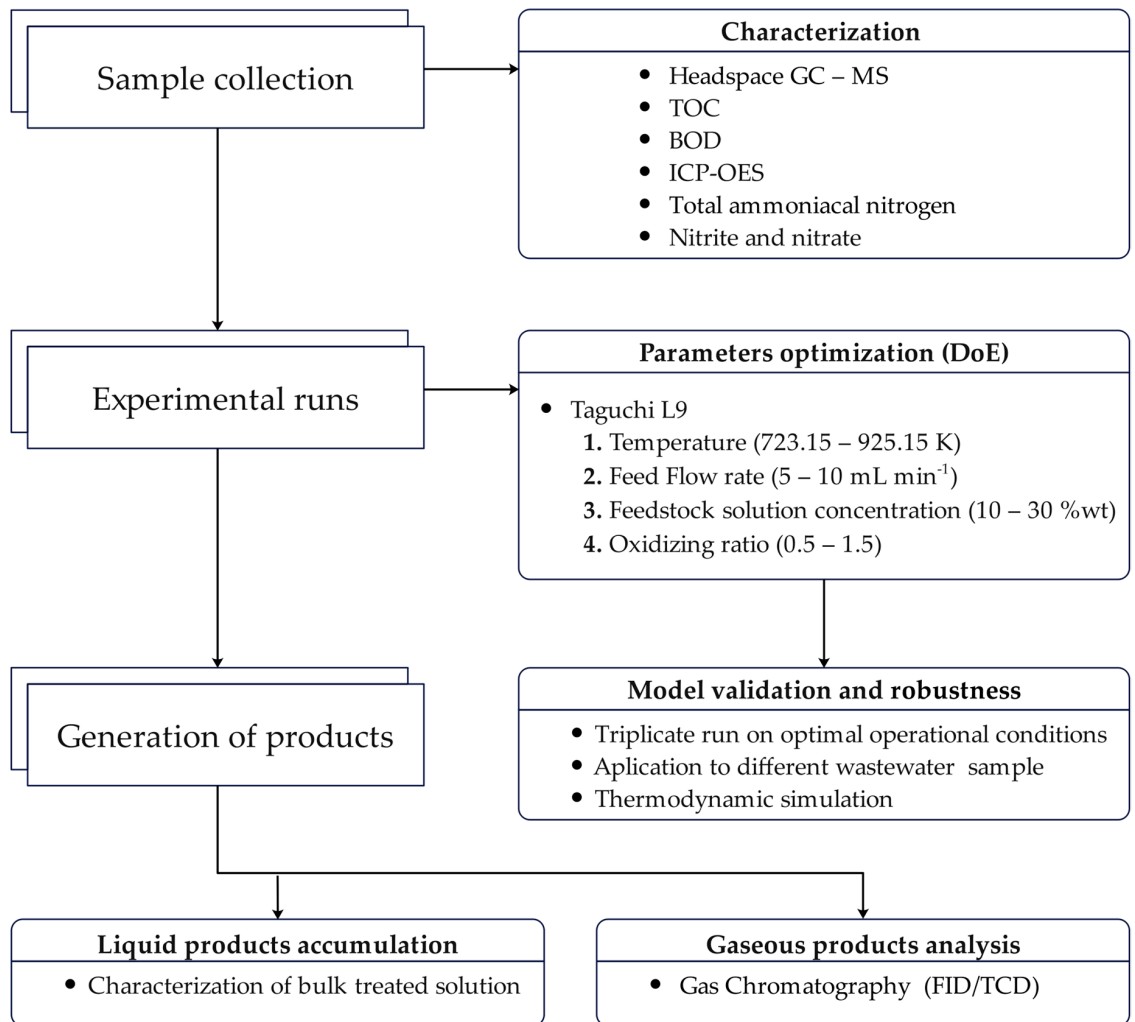

**Figure 2.** Graphic description of the research methodology.

*2.3. Statistical Analysis*

Due to the reduced number of experiments, a Taguchi L9 orthogonal array design was selected to evaluate the interaction of four operating independent parameters on the treatment of refractory organic wastewater via supercritical water technology—specifically, reactor temperature, feed flow rate, feedstock-solution concentration, and oxidizing ratio. Table 2 presents each factor at three levels (coded as 1, 2, and 3).

**Table 2.** Factors and levels for the Taguchi L9 orthogonal array design.

| Operational Parameters | Factors and Levels | | |
|---|---|---|---|
| | 1 | 2 | 3 |
| Reactor temperature (K) | 723.15 | 823.15 | 923.15 |
| Feed flow rate (mL min$^{-1}$) | 5 | 7.5 | 10 |
| Feedstock solution concentration (% wt) | 10 | 20 | 30 |
| Oxidizing ratio | 0.5 | 1.0 | 1.5 |

The aqueous solution samples were prepared by mixing distilled water with refractory effluent, based on the proportions established in the experimental design—that is, 10, 20, or 30% by weight. Preliminary tests with the in natura concentrated (undiluted) effluent resulted in unsatisfactory treatment efficiencies. The effect of hydrogen peroxide was evaluated based on the method of Shin et al. [25], in which the oxygen stoichiometric ratio ($RO_2$) is related to the initial concentration of total organic carbon (TOC) and is defined in Equation (2).

$$RO_2 = \frac{[O_2]}{[TOC]_0} \qquad (2)$$

where $[O_2]$ is the oxygen concentration generated from the hydrogen peroxide solution and $[TOC]_0$ is the initial concentration of TOC in the aqueous waste solution of persistent organic pollutants.

In this work, the reduction of TOC ($R_{TOC}$) was selected as the response variable for the independent factors to evaluate the extension of oxidative degradation. The $R_{TOC}$ was calculated according to Equation (3).

$$R_{TOC} = \frac{[TOC]_0 - [TOC]_f}{[TOC]_0} \cdot 100 \qquad (3)$$

where $[TOC]_f$ is the final concentration of TOC after the SCWO treatment process.

*2.4. Characterization and Analytical Methods*

To evaluate the quality of the raw and treated wastewater, the samples were characterized according to selected physical–chemical parameters, before and after the supercritical water oxidation process. For this purpose, analysis was performed following the Standard Methods for the Examination of Water and Wastewater [26].

The TOC analyses were performed using a total organic carbon analyzer (TOC–L, Shimadzu, Kyoto, Japan), and the total organic carbon concentration was calculated by the difference between the total carbon (TC) and the inorganic carbon (IC). The pH was determined by potentiometry. Nitrite, nitrate, and chemical oxygen demand (CDO) quantifications were conducted by ultraviolet visible spectroscopy (UV–VIS, PerkinElmer 365, Waltham, MA, USA). Biological oxygen demand (BOD) analyses were performed using an optical oximeter model HQ40D from Hach in a BOD incubator supplied by Tecnal, Piracicaba, SP, Brazil.

The reduction of COD was calculated according to Equation (4).

$$R_{COD} = \frac{[COD]_0 - [COD]_f}{[COD]_0} \cdot 100 \qquad (4)$$

where $[COD]_0$ and $[COD]_f$ are the initial and final concentration of COD, respectively, in the inlet and outlet of the SCWO reactor.

The reduction of BOD was calculated according to Equation (5).

$$R_{COD} = \frac{[BOD]_0 - [BOD]_f}{[BOD]_0} \cdot 100 \qquad (5)$$

where $[BOD]_0$ and $[BOD]_f$ are the initial and final concentrations of BOD, respectively, in the inlet and outlet of the SCWO reactor.

The presence and concentration of metallic species were analyzed by inductively coupled plasma optical emission spectroscopy (ICP–OES, Perkin Elmer 7300 DV, Waltham, MA, USA). Total ammoniacal nitrogen (N–NH$_3$) was determined by potentiometry, using an ammonium ion-selective electrode and a multiparameter meter. The N–NH$_3$ corresponds to the sum of the ionized ammonia ($NH_4^+$) and unionized ammonia ($NH_3$).

Samples were also characterized by headspace gas chromatography coupled with mass spectrometry (headspace GC–MS, 7890A, Agilent, Santa Clara, CA, USA). Before

injection, the sample was incubated at 358.15 K for 20 min and agitated at 500 rpm at time intervals of 1 min, followed by rests of 0.5 min. The injection syringe temperature was kept at 373.15 K and the sample volume was 750 μL. A HP-5MS column (30 m × 0.25 mm × 0.25 μm), supplied by Agilent, Santa Clara, CA, USA, was used. The chromatograph oven was held at an initial temperature of 313.15 K for 5 min, followed by a temperature ramp of 278.15 K min$^{-1}$ until a final temperature of 333.15 K was reached and, then, a second temperature ramp of 283.15 min$^{-1}$ until a final temperature of 423.15 K was reached. Finally, a third temperature ramp of 323.15 K min$^{-1}$ took the temperature up to the final temperature of 523.15 K, which was held until the end of the analysis.

The composition of the gas generated during the degradation of the contaminants present in the target effluent was determined by gas chromatography (ThermoFinnigan TRACE GC, Waltham, MA, USA) equipped with a thermal conductivity and flame ionization detectors (TCD and FID), a 10-way valve system, and a Porapak column in series with a 13X Molecular Sieve. The analyses were conducted in isothermal mode, with the columns at 328.15 K and the detector at 403.15 K. Argon was used as the carrier gas at a constant flow rate. A standard gas mixture with the composition ($v/v$) of $H_2$ (50.01%), $CO_2$ (2.04%), $C_2H_4$ (9.95%), $C_2H_6$ (10.02%), $N_2$ (21.11%), $CH_4$ (4.86%), and $CO$ (2.01%) was used in the equipment calibration.

### 2.5. Thermodynamic Simulation: Gibbs Energy Minimization Model

The equilibrium composition can be determined for a system with multiple components and phases at conditions of constant pressure and temperature by directly minimizing the global Gibbs energy of the system. Considering the number of moles of each component in each phase, Equation (6) represents this equation for a system composed of a gas, a liquid, and a solid phase [27].

$$min\ G = \sum_{i=1}^{NC} n_i^g \mu_i^g + \sum_{i=1}^{NC} n_i^l \mu_i^l + \sum_{i=1}^{NC} n_i^s \mu_i^s \tag{6}$$

The model was developed considering the following restrictions: non-negativity of the number of moles represented by Equation (7) of each component in each phase and the balance of moles obtained by the atomic balance for reactive systems, represented by Equation (8).

$$n_i^g, n_i^l, n_i^s \geq 0 \tag{7}$$

$$\sum_{i=1}^{NC} a_{mi}(n_i^g + n_i^l + n_i^S) = \sum_{i=1}^{NC} a_{mi} n_i^0, \ m = 1, \ldots, NE \tag{8}$$

where $g$, $l$, and $s$ represent the gas, liquid, and solid phases; $n_i$ and $a_{mi}$ are the number of moles for each component and the atom of each element in a molecule. $NC$ and $NE$ are the number of components and types of atoms in the system, in that order. The overall Gibbs energy has been minimized by considering that the components are only in the gas phase; solid carbon was considered a possible compound and represented in the solid phase as a pure component, C(s). Equation (9) represents the Gibbs energy equation, including these considerations.

$$G = \sum_{i=1}^{NC} n_i^g \left( \mu_i^g + RT(\ln P + \ln y_i + \ln \phi_i) \right) \tag{9}$$

Non-ideality is represented by the fugacity coefficient, calculated by the virial equations of state truncated at the second coefficient. The second virial coefficient was calculated

using the Pitzer correlation (Kenneth Pitzer et al. [28]), modified by [29]. The calculation of the fugacity coefficient is provided by Equation (10).

$$\ln \hat{\phi}_i = \left[ 2\sum_{j}^{m} y_j B_{ij} - B \right] \frac{P}{RT} \tag{10}$$

Since $\mu_i^g$ and $y_i$ are the chemical potential and the mole fraction of the component, $R$ is the gas constant, $T$ is the temperature of the system, $P$ the pressure, $\phi_i$ and $\hat{\phi}_i$ are the fugacity coefficients of the pure component and in the mixture, respectively, and $m$ is the atom in a molecule; $B$ is the second coefficient of the virial and $B_{ij}$ is this cross coefficient.

The fugacity coefficient of the mixture is calculated using the virial equation of state. The presence of 19 chemical compounds in the effluent of the reaction system was considered. These compounds were selected from the experimental and modeling work for similar systems reported in the literature [30,31]. Table 3 identifies the selected compounds and their thermodynamic/critical properties.

**Table 3.** Considered compounds during simulations and their thermodynamic/critical properties [32].

| Compound | Chemical Formula | $T_C$ (K) | $P_C$ (MPa) | $V_C$ (m$^3$/kmol) | $\omega$ (−) |
|---|---|---|---|---|---|
| Chloroform | $CHCl_3$ | 536.4 | 5.47 | 0.024 | 0.218 |
| Methanol | $CH_3OH$ | 512.6 | 8.09 | 0.012 | 0.556 |
| Acetonitrile | $C_2H_3N$ | 545.5 | 4.83 | 0.017 | 0.278 |
| Water | $H_2O$ | 647.3 | 22.10 | 0.056 | 0.348 |
| Hydrogen | $H_2$ | 33.0 | 1.30 | 0.064 | 0.000 |
| Ethane | $C_2H_6$ | 305.4 | 4.82 | 0.148 | 0.105 |
| Propane | $C_3H_8$ | 369.9 | 4.20 | 0.200 | 0.152 |
| Ethylene | $C_2H_4$ | 283.1 | 5.05 | 0.124 | 0.073 |
| Propylene | $C_3H_6$ | 369.9 | 4.54 | 0.182 | 0.143 |
| Carbon monoxide | CO | 133.0 | 3.50 | 0.093 | 0.041 |
| Carbon dioxide | $CO_2$ | 304.2 | 7.39 | 0.094 | 0.420 |
| Methane | $CH_4$ | 191.1 | 4.58 | 0.099 | 0.013 |
| Ammonia | $NH_3$ | 405.6 | 11.35 | 0.072 | 0.250 |
| Nitric oxide | NO | 180.0 | 6.48 | 0.058 | 0.607 |
| Nitrogen dioxide | $NO_2$ | 431.0 | 10.10 | 0.169 | 0.860 |
| Nitrogen | $N_2$ | 126.2 | 3.39 | 0.086 | 0.040 |
| Hydrogen chloride | HCl | 324.7 | 8.31 | 0.081 | 0.133 |
| Chlorine | $Cl_2$ | 416.9 | 7.99 | 0.124 | 0.090 |
| Hydrogen peroxide | $H_2O_2$ | 728.0 | 22.00 | 0.073 | 0.359 |

The thermodynamic properties required to conduct thermodynamic analysis of the reaction system, including heat capacity, enthalpy, and Gibbs energy of formation, were obtained from the literature [32]. Table 4 shows the feed operating conditions (% wt of reactants, pressure, and temperature range) required for thermodynamic analysis of the reaction system that were selected, based on the characterization of the laboratory wastewater and with slight extrapolation of the operational limitations of the experimental setup.

**Table 4.** Experimental operating conditions used in the SCWO thermodynamic analysis via Gibbs energy minimization.

| | Min | Max | Unit |
|---|---|---|---|
| $CHCl_3$ | 5 | 10 | % wt |
| $CH_3OH$ | 5 | 10 | % wt |
| $C_2H_3N$ | 5 | 10 | % wt |
| $H_2O$ | 60 | 80 | % wt |
| $H_2O_2$ | 5 | 10 | % wt |
| Temperature | 673.15 | 1073.15 | K |
| Pressure | 25 | 25 | MPa |

# 3. Results and Discussion

## 3.1. Supercritical Water Oxidation Treatment and Organic Degradation

The chromatogram of the untreated laboratory wastewater sample is shown in Figure 3. The results showed that the effluent was mainly composed of methanol and acetonitrile, two organic substances widely used as the mobile phase in HPLC analysis, and chloroform, used as a solvent for extraction.

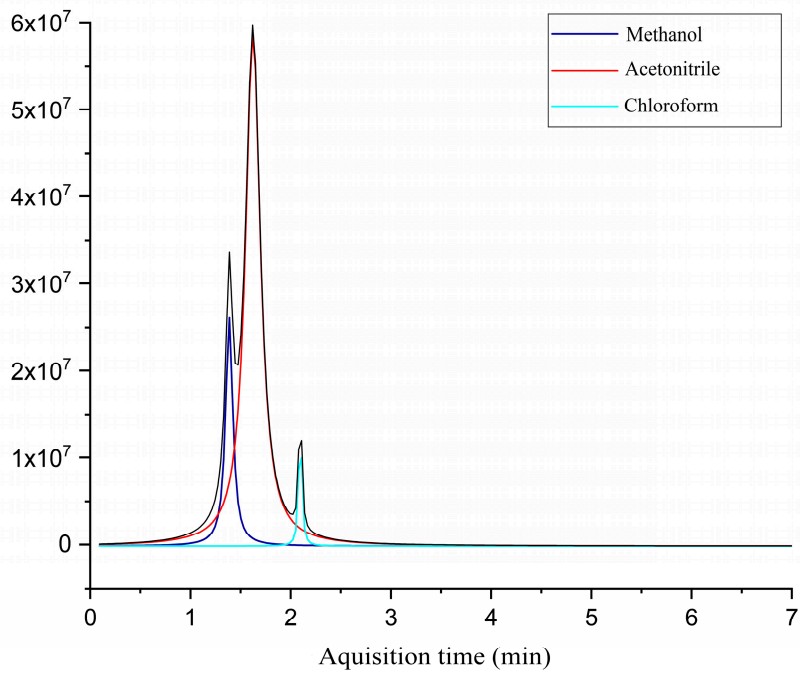

**Figure 3.** Chromatogram of refractory organic compounds identified in the wastewater samples obtained by GC-MS analysis.

To ensure that the system remained under supercritical conditions, the preheater temperature and the system pressure were maintained, respectively, at constant values of 623.15 K and 25 MPa. The safety limits of the experimental apparatus were also taken into consideration for the definitions of the temperature and the working pressure. The volumetric flow and reactor temperature ranges were defined based on the limitations of the isocratic pump model (maximum of 10 mL min$^{-1}$) and the set of split furnaces and infrared ceramic heaters (maximum of 973.15 K), respectively.

The main effects of each selected operational parameter on the degradation of the wastewater samples via SCWO was evaluated in terms of reduction of total organic carbon ($R_{TOC}$). The obtained results are depicted in Figure 4.

Among the four evaluated factors, it was observed that both the temperature and the $RO_2$ showed a greater influence on the $R_{TOC}$ response variable. On the other hand, the feed flow rate and the effluent concentration slightly influenced the TOC reduction. In general, except for temperature, a greater reduction in TOC values and, consequently, a greater degradation of organic compounds were achieved at the highest levels of all operational parameters evaluated.

In the experiments carried out at a temperature of 923.15 K and in the presence of oxygen peroxide as an oxidizing agent, the occurrence of gas accumulation was observed, which resulted in instability of the system and momentary reductions in the system pressure. This fact may have disfavored the degradation of the organic contaminants in comparison with the experiments at the temperature of 823.15 K and resulted in a lower $R_{TOC}$. The obtained results are summarized in Table 5.

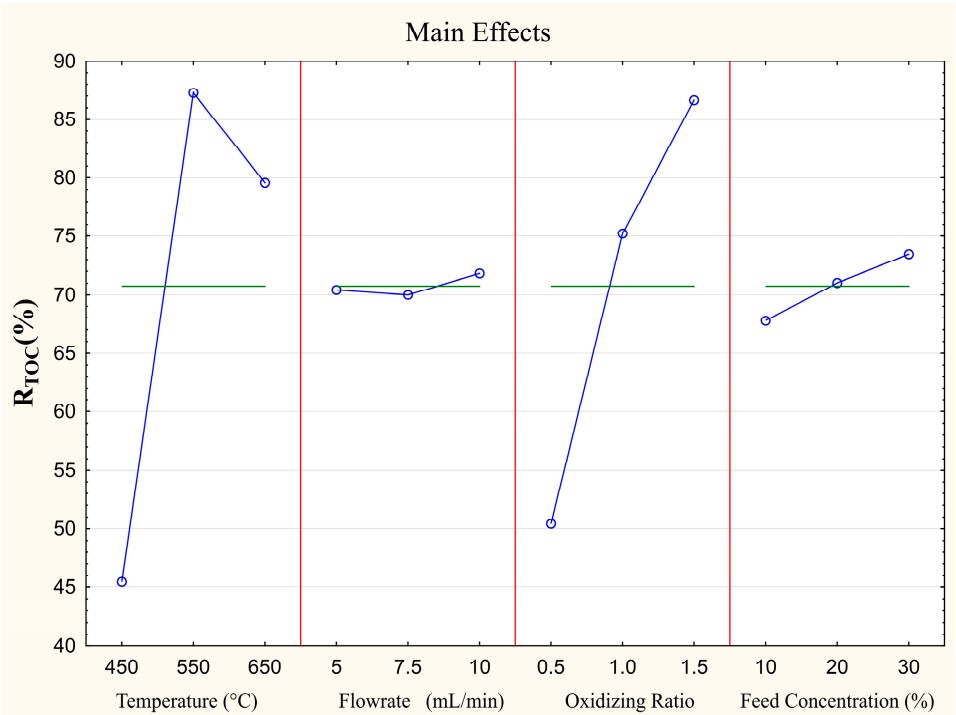

**Figure 4.** Main effects of the operational parameter in the treatment of refractory organic pollutants via SCWO.

**Table 5.** Reduction of TOC ($R_{TOC}$) in the treatment of refractory organic pollutants via supercritical water oxidation.

| Run | Temperature (K) | Feed Flow Rate (mL min$^{-1}$) | Oxidizing Ratio - | Solution Concentration (% wt) | $R_{TOC}$ (%) |
|---|---|---|---|---|---|
| 1 | 723.15 | 5 | 0.5 | 10 | 21.8 |
| 2 | 723.15 | 7.5 | 1.0 | 20 | 49.4 |
| 3 | 723.15 | 10 | 1.5 | 30 | 65.1 |
| 4 | 823.15 | 5 | 1 | 30 | 94.1 |
| 5 | 823.15 | 7.5 | 1.5 | 10 | 99.4 |
| 6 | 823.15 | 10 | 0.5 | 20 | 68.3 |
| 7 | 923.15 | 5 | 1.5 | 20 | 95.4 |
| 8 | 923.15 | 7.5 | 0.5 | 30 | 61.2 |
| 9 | 923.15 | 10 | 1 | 10 | 82.1 |

According to García-Jarana et al. [33], it is important to consider the effect of oxygen excess on the oxidation of nitrogen-containing compounds in wastewater. The incomplete oxidation of organic nitrogen-containing compounds can result in the formation of ammonia, a recalcitrant compound. In general, the formation of ammonia and/or nitrates as by-products of the treatment of effluents contaminated by nitrogenous substances is observed when the concentration of oxygen available in the medium is below the stoichiometric value.

On the other hand, the influence of the concentration of organic compounds in the feed solution was evaluated by Ma et al. [34]. The results indicated that increasing the aniline concentration from 0.5 to 1.0% wt increased the $R_{TOC}$ efficiency by 7.4%. However, a further increase in aniline concentration resulted in only a slight increase in removal efficiency, and the influence of this parameter was negligible from that point on [34].

In this study, according to Taguchi's optimization, the highest $R_{TOC}$ could be achieved at a temperature of 823.15 K, a feed rate of 10 mL min$^{-1}$, an oxidizing ratio of 1.5, and a sample concentration of 30%. To validate the experimental design and confirm the

robustness of the methodology employed, three additional experiments were conducted under the optimized conditions and an average $R_{TOC}$ of 99.9% $\pm$ 0.1 was achieved.

Regarding reaction mechanisms involving organic matter oxidation/degradation under supercritical water conditions, monitoring such reactions experimentally is difficult, due to extreme supercritical reaction conditions. Literature-based mechanisms are promoted by radical species, which occur by initiation, propagation, and termination stages. Equations (11)–(14) show some mechanism for radical species formation.

$$H_2O \rightarrow H \bullet + OH \bullet \tag{11}$$

$$H_2O_2 \rightarrow 2OH \bullet \tag{12}$$

$$2\,H_2O_2 \rightarrow 2\,H_2O + O_2 \tag{13}$$

$$H \bullet + O_2 \rightarrow HO_2 \bullet \tag{14}$$

According to Li et al. [35], when $H_2O_2$ is used as an oxidant, the hydrogen abstraction phase occurs according to the reactions presented in Equations (15)–(17) [35].

$$OH\bullet + H_2O_2 \rightarrow H_2O + HO_2 \bullet \tag{15}$$

$$OH \bullet + HO_2 \bullet \rightarrow H_2O + O_2 \tag{16}$$

$$HO_2 \bullet + HO_2 \bullet \rightarrow H_2O_2 + O_2 \tag{17}$$

Then, in the propagation phase, hydrogen, hydroxyl, and hydroperoxyl radicals decompose organic compounds into new radicals, as shown in Equations (18)–(21).

$$R \bullet + O_2 \rightarrow RO_2 \tag{18}$$

$$RO_2 \bullet + RH \rightarrow ROOH + R \bullet \tag{19}$$

$$RO_2 \bullet \rightarrow HOOR \tag{20}$$

$$R \bullet \rightarrow R \bullet + C \rightarrow RH \tag{21}$$

Finally, in the termination stage, free radicals interact to generate novel compounds, usually resulting in the formation of species characterized by simple molecular structures, as evidenced in Equations (22)–(25).

$$R \bullet + R \bullet \rightarrow R - R \tag{22}$$

$$R \bullet + RO \bullet \rightarrow ROR \tag{23}$$

$$RO \bullet + RO \bullet \rightarrow ROOR \tag{24}$$

$$R \bullet + ROO \bullet \rightarrow ROOR \tag{25}$$

The gas production during supercritical water processes is characterized by its inherent complexity, involving a succession of physical transformations and numerous chemical reactions occurring within the reactor, as noted by Hantoko et al. [36]. In summary, the comprehensive gasification process is represented by the overall reaction depicted in Equation (26).

$$CHxOy + (2 - y)H_2O \rightarrow CO_2 + (2 - y + x/2)H_2 \tag{26}$$

### 3.2. Liquid Phase Characterization and Analysis

The physical–chemical parameters of the raw and treated samples via SCWO under optimized conditions (temperature of 823.15 K, a feed flow rate of 10 mL min$^{-1}$, and an oxidizing ratio of 1.5) are shown in Table 6. In accordance with the $R_{TOC}$ results, elevated levels of reduction for both chemical oxygen demand (COD) and biochemical

oxygen demand (BOD) of 99.2% and 91.5% were observed, respectively. Additionally, a reduction in the concentration of metals and inorganic substances (such as sodium, magnesium, and sulfur), due to the decrease in the solubility of these compounds and the formation of insoluble metallic oxides inside the system, especially in the intersection between the preheater and the tubular reactor, was observed. This was possible due to the distinct attributes exhibited by water under supercritical conditions. Within this context, there was a notable reduction in the dielectric constant ($\varepsilon$) and ionic product (Kw) values, consequently leading to a considerable diminution in the solubility of inorganic substances, including oxides and salts, which may explain the removal of metals observed after SCWO treatment [37].

**Table 6.** Physical–chemical parameters of the raw and treated wastewater samples under optimized SCWO conditions.

| Parameters [1] | Sample | | | Reduction (%) |
|---|---|---|---|---|
| | Raw | Treated [2] | Uncertainty [3] | |
| TOC | 47,980 ± 116 | 60.1 ± 0.24 | - | 99.9 |
| COD | 204,876 | 17,458 | 0.060 | 91.5 |
| BOD | 489,056 | 4162.5 | 0.145 | 99.2 |
| Nitrite | 0 | 0 | 0.030 | - |
| Nitrate | 0 | 0.2 | 0.004 | - |
| N–NH$_3$ | 12.3 | 362.8 | - | - |
| Aluminum (Al) | 0.03 | 0.01 | 0.0023 | 66.7 |
| Calcium (Ca) | 2.3 | 1.3 | 0.003 | 43.5 |
| Chromium (Cr) | < | 0.4 | 0.003 | - |
| Iron (Fe) | 0.02 | < | 0.003 | - |
| Potassium (K) | 7.4 | 0.6 | 0.004 | 91.9 |
| Magnesium (Mg) | 0.5 | 0.2 | 0.001 | 60 |
| Sodium (Na) | 11.5 | 2.75 | 0.005 | 76.1 |
| Nickel (Ni) | 0.004 | 0.05 | 0.004 | - |
| Sulfur (S) | 1152.8 | 21.5 | 0.0002 | 98.1 |
| Zinc (Zn) | 0.08 | < | 0.006 | - |

Notes: [1] Treatment conditions: temperature (823.15 K), feed flow rate (10 mL min$^{-1}$), and H$_2$O$_2$ ratio of 1.5. [2] Uncertainty = expanded uncertainty (U), which is based on the combined standard uncertainty, with a 95% confidence level. [3] Regulated limit values are expressed in mg L$^{-1}$.

Nevertheless, an increase in the concentration of metallic species, mainly nickel (0.004 mg L$^{-1}$ → 0.05 mg L$^{-1}$) and chromium (0.4 mg L$^{-1}$), resulting from the leaching of the inner walls of the tubular reactor (Inconel 625 alloy: ~61% nickel, ~22% chromium, and ~9% molybdenum) due to the corrosive power of water under supercritical conditions, was also observed. Additionally, the concentration of nitrate and ammoniacal nitrogen (N–NH$_3$) also increased at the end of the process. In general, it is expected that the presence of nitrite is the result of an incomplete oxidation reaction and, therefore, as an excess of oxygen was used during the treatment, this was not found in the medium, indicating that the acetonitrile was completely converted into nitrate (0.2 mg L$^{-1}$) and ammonia.

Yang et al. [38] evaluated the use of SCWO in the decomposition of 44 nitrogenous compounds and achieved efficiencies greater than 80%, in terms of TOC removal, for all compounds evaluated at temperatures up to 823.15 K. The researchers identified gaseous nitrogen, organic nitrogen, ammoniacal nitrogen, and nitrate as the main nitrogen-containing compounds after treatment. In accordance with the results available in the literature, ammoniacal nitrogen (12.3 mg L$^{-1}$ → 362.8 mg L$^{-1}$) was observed as the main by-product of the treatment of organic nitrogenous compounds via oxidation in supercritical water. According to Bermejo et al. [39], temperatures above 973.15 K (higher than the Inconel 625 rating at the typical working pressures for SCWO) are necessary for the degradation of ammonia via oxidation in supercritical water, reaching up to 1073.15 K for an ammonia concentration of 7% wt. Additionally, another strategy that could be used for the enhance-

ment of the ammonium removal during the SCWO process is the use of organic solvents, such as isopropanol, as co-fuels [40].

The headspace GC–MS analysis showed that, even at optimal conditions, acetonitrile was not completely oxidized, as shown in Figure 5. The 2-amino-1-propanol was identified as the predominant compound. The coexistence of amino and hydroxyl functional groups on the same molecule, both acting as hydrogen bond donors and acceptors, leads to many possible hydrogen bonding interactions. This effect is especially significant when these groups are closer to each other [41]. Additionally, the complete degradation of both chloroform and methanol was achieved. The presence of ammoniacal nitrogen in the reaction medium may have contributed to this result, since, according to Shimoda et al. [42], the methanol conversion during the supercritical water oxidation process of ammonia/methanol was higher than that of isolated methanol.

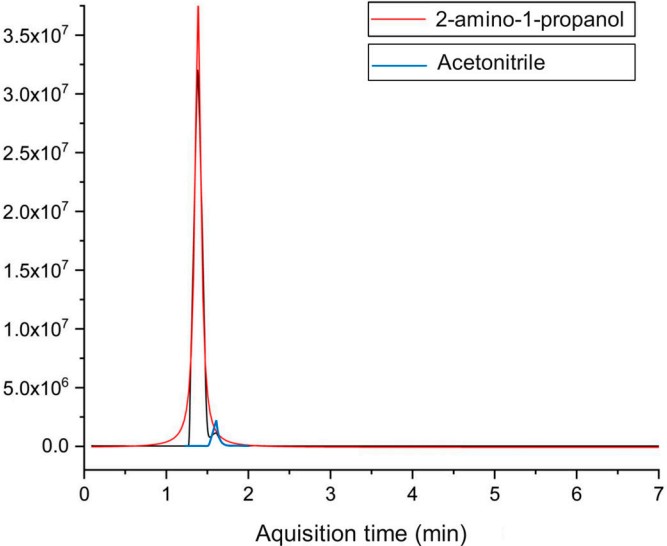

**Figure 5.** Detailed chromatogram of the organic byproducts identified via headspace GC–MS on the sample treated via supercritical water oxidation under optimized conditions (temperature of 823.15 K, feed rate of 10 mL min$^{-1}$, oxidizing ratio of 1.5).

### 3.3. Gaseous Products Analysis

The analysis of the gases produced during the SCWO treatment was conducted by gas chromatography. At all evaluated temperatures, it was observed that, among the identified gases, the production of carbon dioxide ($CO_2$), followed by methane ($CH_4$), was predominant. Specifically, at the lowest temperature (723.15 K), the fraction of $CO_2$ was significantly higher than the $CH_4$ concentration. The presence of carbon monoxide in the produced gases was observed below 6% in all evaluated conditions, while the average of hydrogen concentration was below 10%.

Under optimized conditions (temperature of 823.15 K, feed flow rate of 10 mL min$^{-1}$, oxidizing ratio of 1.5), the gaseous product composition was 68.1% $CO_2$, 23.3% $CH_4$ and 8.6% $H_2$. At higher temperatures, an increase in the hydrogen fraction and a decrease in the fraction of methane were observed. This suggests that methane may have been consumed as a reactant in other reactions, resulting in the increase observed for the hydrogen concentration. Additionally, the CO observed in small concentration in the experimental runs conducted with an oxidizing ratio equal to 0.5 was, for the most part, completely oxidized into $CO_2$ when the concentration of oxidizing agent was increased.

In line with the obtained results, Liu et al. [43] observed that, during the partial oxidation of indole ($C_8H_7N$) in supercritical water, the $CO_2$ yield increased rapidly with the increase of the oxidizing ratio, while the $H_2$ and $CH_4$ concentrations decreased. Additionally, the results obtained are also corroborated by the study reported by Benjamin and Savage, who observed that, as the residence time increases, the maximum CO yield

decreases, resulting in a higher $CO_2$ fraction and suggesting that CO is an intermediate for the formation of $CO_2$ [44]. At higher temperatures, CO oxidation occurs more rapidly, resulting in higher $CO_2$ yields. Thus, given an adequate reaction time, oxidizing ratio, and temperature, it is expected that all the carbon fed to the reactor ends up being completely converted into $CO_2$ [45].

According to Chakinala et al. [46], methanol is considered a stable compound at temperatures below 873.15 K, in the absence of a catalyst. The decomposition of methanol can occur through different routes. One route involves the abstraction of a hydrogen atom from an oxygen atom, resulting in the generation of an $H_3CO \bullet$ radical. This radical can decompose into formaldehyde and release an $H \bullet$ radical. Then, formaldehyde decomposes directly into CO and $H_2$, or, intermediately, it can oxidize into formic acid, which decomposes into $CO_2$ and $H_2$. Another possible initiation route is the hydrogen abstraction on the α-carbon atom, which can also lead to the formation of formaldehyde intermediates. Cleavage of the C–O bond is a dehydration pathway that occurs when an $H \bullet$ radical reacts with an OH group present in the compound, forming water and a $CH_3 \bullet$ radical, which can then lead to the formation of methane.

The distribution of gaseous products indicates that nitrogenous compounds are difficult to be completely oxidized to $N_2$, producing favorably recalcitrant intermediates instead (N–$NH_3$), as discussed earlier. Under appropriate conditions, such as higher temperatures, it is possible to detect $N_2$ in the gaseous product. However, in the tests performed, on average, the fraction of $N_2$ observed was around 4.4%. Crain et al. [47] evaluated different pyridine degradation pathways; however, $N_2$ was not detected in the gaseous state, with only traces of $NO_2^-$ present. Furthermore, the degradation of complex organic compounds containing nitrogen was evaluated by Al-Duri et al. [48] During the supercritical water oxidation of 1,8-diazabicyclo [5.4.0] undec-7-ene (commonly known as DBU), it was observed that the organic nitrogen was mainly converted into N–$NH_3$ and an insignificant amount of $NO_2^-$ and $NO_3^-$ was produced. At higher temperatures, the formation of gaseous $N_2$ was observed; however, most of the organic nitrogen remained in the form of N–$NH_3$ in the liquid effluent, even at 923.15 K, suggesting that N–$NH_3$ could not be effectively decomposed at lower temperatures.

### 3.4. Wastewater Generated at the Federal University of Goiás–UFG

To evaluate the robustness of the methodology developed for the degradation of refractory organic compounds, the supercritical water oxidation process was applied for the treatment of the wastewater generated at the analytical instrumentation laboratory located at the Institute of Chemistry of the Federal University of Goiás–UFG (Brazil). It was observed that, due to the specialization of this laboratory in HPLC analysis, the composition of the wastewater was very similar to the original target effluent, consisting mainly of acetonitrile and methanol, as shown in Figure 6.

As stated before, the treatment efficiency was evaluated in terms of the degradation of TOC. A three-hour experiment was conducted at the optimal conditions for temperature, feed flow rate, and feed solution concentration (823.15 K, 10 mL min$^{-1}$, and 30%, respectively) and a higher excess of oxidizing agent ($n = 2$). The removal of TOC was 99.5%, showing the high/adequate efficiency of the proposed procedure in the treatment of this type of wastewater. Although there were no significant changes in the TOC removal, due to the higher oxidizing ratio, a change in the treatment byproducts was observed. The head-space GC–MS analysis showed that the organic compounds were completely oxidized, including the acetonitrile, and only 2-amino-1-propanol was identified, as illustrated in Figure 7.

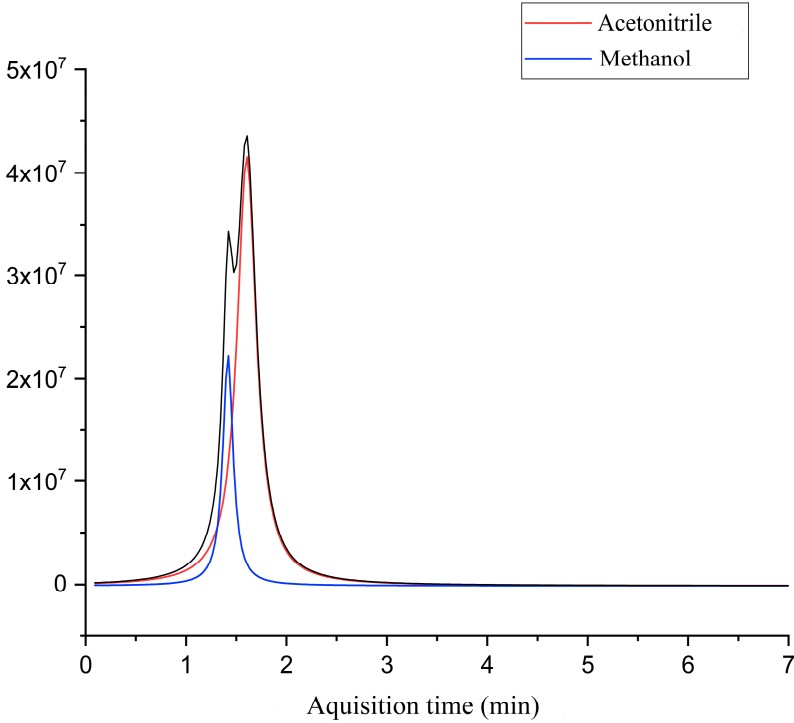

**Figure 6.** Chromatogram of refractory organic compounds found in the wastewater samples obtained at the Federal University of Goiás–UFG, Brazil.

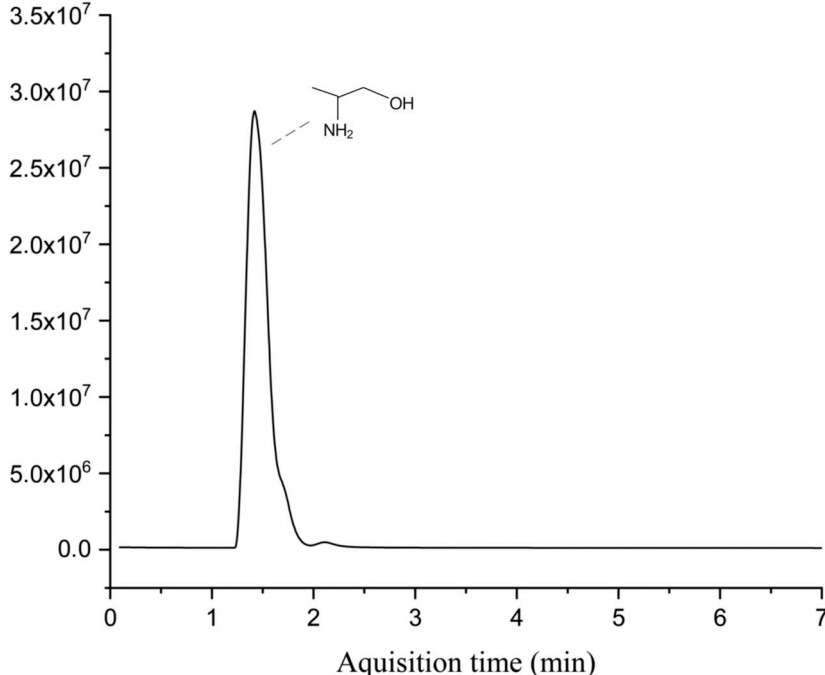

**Figure 7.** Detailed chromatogram of the organic by-product identified via headspace GC–MS on the sample treated via SCWO.

### 3.5. Thermodynamic Simulation Data

The Gibbs energy minimization methodology was used for the thermodynamic analysis of the SCWO treatment of laboratory wastewater under isothermal conditions. Figure 8 shows the correlation matrix of the simulated data, with correlations estimated by the Spearman method. It was observed that increasing the temperature favors the formation

of hydrogen, indicating the occurrence of endothermic reactions such as the water–gas shift reaction. Methane negatively correlates with temperature, suggesting that it forms hydrogen and carbon monoxide. The addition of methanol and acetonitrile promotes the formation of hydrogen, although with a correlation below 0.09. Temperature is the main factor for hydrogen formation, in line with studies by Withag et al. [49], Yan et al. [50], Fedyaeva and Vostrikov [30], and Aslam and Twaiq [31].

| | T (°C) | CHCl₃ (in) | CH₃OH (in) | C₂H₃N (in) | H₂O (in) | H₂O₂ (in) | H₂O | H₂ | CO | CO₂ | NH₃ | HCl | CH₄ | N₂ | | |
|---|---|---|---|---|---|---|---|---|---|---|---|---|---|---|---|---|
| **T (°C)** | 1.000 | 0.000 | 0.000 | 0.000 | 0.000 | 0.000 | 0.211 | −0.346 | 0.622 | −0.440 | 0.022 | 0.000 | −0.371 | −0.059 | | 1.000 |
| **CHCl₃ (in)** | 0.000 | 1.000 | −0.201 | 0.020 | −0.310 | 0.000 | −0.083 | −0.167 | −0.055 | 0.044 | 0.371 | 0.845 | −0.372 | −0.292 | | 0.846 |
| **CH₃OH (in)** | 0.000 | −0.201 | 1.000 | −0.201 | −0.424 | 0.456 | −0.149 | 0.359 | 0.196 | 0.211 | −0.462 | −0.309 | 0.394 | −0.100 | | 0.692 |
| **C₂H₃N (in)** | 0.000 | 0.020 | −0.201 | 1.000 | −0.310 | 0.000 | −0.784 | 0.571 | 0.581 | 0.698 | 0.844 | −0.101 | 0.784 | 0.996 | | 0.538 |
| **H₂O (in)** | 0.000 | −0.310 | −0.424 | −0.310 | 1.000 | −0.775 | 0.646 | −0.563 | −0.450 | −0.646 | −0.392 | −0.131 | −0.340 | −0.255 | | 0.385 |
| **H₂O₂ (in)** | 0.000 | 0.000 | 0.456 | 0.000 | −0.775 | 1.000 | −0.167 | 0.160 | 0.090 | 0.274 | 0.169 | 0.169 | −0.434 | −0.345 | | 0.231 |
| **H₂O** | 0.211 | −0.083 | −0.149 | −0.784 | 0.646 | −0.167 | 1.000 | −0.910 | −0.599 | −0.946 | −0.576 | 0.265 | −0.503 | −0.688 | | 0.077 |
| **H₂** | −0.346 | −0.167 | 0.359 | 0.571 | −0.563 | 0.160 | −0.910 | 1.000 | 0.483 | 0.902 | 0.247 | −0.524 | −0.280 | 0.017 | | −0.077 |
| **CO** | 0.622 | −0.055 | 0.196 | 0.581 | −0.450 | 0.090 | −0.599 | 0.483 | 1.000 | 0.375 | 0.362 | −0.339 | −0.144 | 0.152 | | −0.231 |
| **CO₂** | −0.440 | 0.044 | 0.211 | 0.698 | −0.646 | 0.274 | −0.946 | 0.902 | 0.375 | 1.000 | 0.516 | −0.224 | 0.600 | 0.746 | | −0.385 |
| **NH₃** | 0.022 | 0.371 | −0.462 | 0.844 | −0.392 | 0.169 | −0.576 | 0.247 | 0.362 | 0.516 | 1.000 | 0.371 | 0.307 | 0.271 | | −0.538 |
| **HCl** | 0.000 | 0.845 | −0.309 | −0.101 | −0.131 | 0.169 | 0.265 | −0.524 | −0.339 | −0.224 | 0.371 | 1.000 | −0.345 | −0.273 | | −0.692 |
| **CH₄** | −0.371 | −0.372 | 0.394 | 0.784 | −0.340 | −0.434 | −0.503 | −0.280 | −0.144 | 0.600 | 0.307 | −0.345 | 1.000 | 0.799 | | −0.846 |
| **N₂** | −0.059 | −0.292 | −0.100 | 0.996 | −0.255 | −0.345 | −0.688 | 0.017 | 0.152 | 0.746 | 0.271 | −0.273 | 0.799 | 1.000 | | −1.000 |

**Figure 8.** Correlation matrix of elements involved in the SCWO reaction system of laboratory wastewater.

The obtained results indicate that adding chloroform to the residue used in the reactor feed stream tends to maximize the formation of HCl. This behavior was expected, considering that both molecules and HCl are compounds derived from chlorine. The possibility of HCl formation over a wide concentration range in SCWO reactions deserves attention, due to the possibility of corrosion on the reactor walls (Wash). However, increasing the temperature and other reagents (methanol, acetonitrile, and hydrogen chloride) minimized the formation of this component. Thus, conditions that maximize hydrogen formation tend to minimize HCl formation.

Similar HCl formation results were reported by Aslam and Twaiq [27]. Fedyaeva and Vostrikov [26] showed that there is practically no $Cl_2$ gas produced during the SCWO process, due to the formation of HCl when $Cl_2$ reacts with water. As temperature proved to be the predominant effect on the formation of hydrogen throughout the SCWO of laboratory wastewater, a more detailed study about this thermodynamic effect was conducted and the results are depicted in Figure 9.

Figure 9 shows the obtained results of the thermodynamic analysis of the highest hydrogen production rate for the SCWO reactor feed solution containing 5% of chloroform, 15% of methanol, 15% of acetonitrile, 5% of hydrogen peroxide and 60% of water on a mass basis.

In Figure 9a, the thermodynamic analysis was conducted considering the possibility of the formation of all selected compounds. On the other hand, in Figure 9b, a possible inhibition of methane formation during the SCWO reaction was considered. From the results depicted in both Figure 9 and the correlation matrix, as well as the experimental results, it was observed that increasing the temperature favors the formation of hydrogen and minimizes methane formation. In addition, the amount of water decreased with increasing temperature, due to the formation of hydrogen. Thus, water plays an essential role in this system, acting as a solvent in the reaction medium and a relevant reactant during the SCWO reaction. Although the other components were formed in small quantities,

carbon monoxide and carbon dioxide formation, in accordance with the experimental results, were still considerable under all the conditions evaluated.

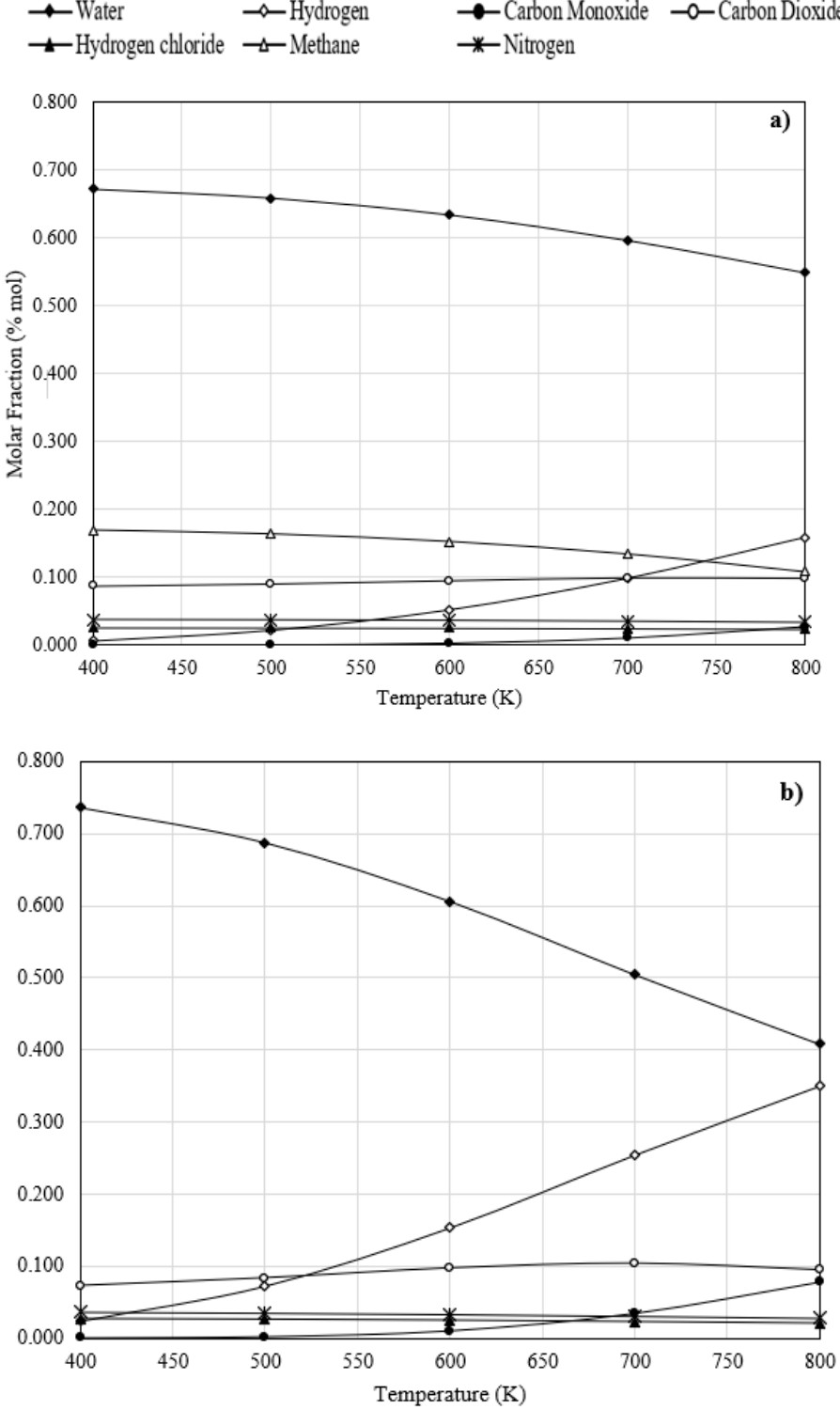

**Figure 9.** Molar compositions as a function of temperature for the SCWO treatment of laboratory wastewater at 25 MPa: (**a**) with formation of methane and (**b**) with methane inhibition.

## 4. Conclusions

This study proposed and developed a treatment process for refractory organic pollutants generated in research and teaching laboratories. A Taguchi L9 orthogonal array design

was selected to analyze the effects of four different operating parameters—temperature, feed flow rate, oxidizing ratio, and feedstock solution concentration—in the supercritical water oxidation process. At optimized conditions—that is, a temperature of 823.15 K, a feed flow rate of 10 mL min$^{-1}$, and an oxidizing ratio of 1.5—the proposed treatment achieved, respectively, a TOC, COD, and BOD reduction of 99.9%, 91.5%, and 99.2% in a reduced time. Also, only $CO_2$, methane, and hydrogen were identified. The formation of ammoniacal nitrogen was found to be a limiting step in the SCWO treatment of refractory organic pollutants containing nitrogen. The SCWO treatment process's robustness was confirmed by successfully applying the developed methodology for the treatment of the wastewater generated at another analytical instrumentation laboratory ($R_{TOC}$ = 99.5%). Additionally, the thermodynamic analysis of the SCWO treatment of laboratory wastewater under isothermal conditions was performed using the Gibbs energy minimization methodology. In accordance with the experimental results, it was observed that increasing the temperature favors the generation of hydrogen, while it inhibits methane formation. Finally, carbon monoxide and carbon dioxide formation were considerable under all the conditions evaluated. In summary: (1.) the efficient treatment via SCWO of highly concentrated (TOC ~ 48,000) real wastewater samples was achieved; (2.) The robustness of the methodology was proved by its successful application on wastewater samples generated at two different laboratories. (3.) The formation of harmful gases, such as NO and NOx, were not observed; (4.) The generation of ammoniacal nitrogen during the SCWO process must be investigated; and (5.) The thermodynamic simulation of the SCWO treatment of laboratory wastewater corroborated with the experimental results.

**Author Contributions:** Conceptualization, M.B.P., G.B.M.d.S., C.G.A. and L.C.-F.; methodology, M.B.P., G.B.M.d.S., C.G.A. and L.C.-F.; software, J.M.d.S.-J., A.C.D.d.F. and R.G.; validation, M.B.P., G.B.M.d.S. and I.M.D.; formal analysis, M.B.P., G.B.M.d.S., J.M.d.S.-J., A.C.D.d.F. and, R.G.; investigation, M.B.P., G.B.M.d.S., and I.M.D.; resources, L.C.-F.; writing—original draft preparation, M.B.P., G.B.M.d.S., I.M.D., J.M.d.S.-J., A.C.D.d.F. and R.G.; writing—review and editing, M.B.P., G.B.M.d.S., C.G.A. and L.C.-F.; visualization, J.M.A.-P., C.G.A. and L.C.-F.; supervision, J.M.A.-P., C.G.A. and L.C.-F.; project administration, C.G.A. and L.C.-F.; funding acquisition, L.C.-F. All authors have read and agreed to the published version of the manuscript.

**Funding:** This study was financed in part by Coordenação de Aperfeiçoamento de Pessoal de Nível Superior–Brasil (CAPES)—Finance Code 001.

**Data Availability Statement:** Data are contained within the article.

**Acknowledgments:** The authors would like to thank the following partner laboratories: (i) Laboratório de Análises de Solos, Adubos e Plantas–DAG/UEM, (ii) Central Analítica do Instituto de Química–CAM/UFG/IQ, (iii) Complexo de Centrais de Apoio à Pesquisa–COMCAP/UEM, (iv) Petroleum and Energy from Biomass research group–PEB/UFS, (v) Instituto Cesumar de Ciência, Tecnologia e Inovação–ICETI/UniCesumar, and (vi) Centro Regional para o Desenvolvimento Tecnológico e Inovação–CRTI/UFG.

**Conflicts of Interest:** The authors declare no conflict of interest.

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
