# Peer review of "Continuous Treatment of Refractory Wastewater from Research and Teaching Laboratories via Supercritical Water Oxidation–Experimental Results and Modeling"

_water, doi:10.3390/w15223926_

Round 1
Reviewer 1 Report
Comments and Suggestions for Authors
The article on "Continuous treatment of refractory wastewater from research and teaching laboratories via supercritical water oxidation - Experimental results and modeling" is well written and generally understandable, but the English needs improvement.
- For instance, in the abstract....line 29.. the text should be write as.... TOC, COD and DBO5 reduction 99.9%, 91.5% and 99.2 % were achieved ....time, respectively.
- Page 9, Table 4... the results presented in the table.... Why the hightest RTOC was achieved at 99.4% with 10 % of concentration instead of 30%?
Comments on the Quality of English Language
The quality of English language should be improved.
Author Response
Reviewer #1:
General comment: The article on "Continuous treatment of refractory wastewater from research and teaching laboratories via supercritical water oxidation - Experimental results and modeling" is well written and generally understandable, but the English needs improvement.
Answer: Thank you. We appreciate the good evaluation of our work.
Specific comments:
1) For instance, in the abstract....line 29.. the text should be written as.... TOC, COD and DBO5 reduction 99.9%, 91.5% and 99.2 % were achieved ....time, respectively.
Answer: It was revised.
2) Page 9, Table 4... the results presented in the table.... Why the highest RTOC was achieved at 99.4% with 10 % of concentration instead of 30%?
Answer: It was observed that, among the four evaluated factors, only the temperature and the concentration of H2O2 showed a significant influence on the RTOC response variable. On the other hand, the feed flow rate and the effluent concentration slightly influenced the RTOC. In this sense, it is not possible to evaluate the highest RTOC achieved (during the experimental planning) taking in account only the concentration of the feed solution. Additionally, although the highest RTOC was achieved with feed concentration of 10%, the main effects of the independent variables (as depicted in Figure 3) showed that higher feed concentrations lead to higher RTOC values.
3) The quality of English language should be improved.
Answer: Thanks for the suggestion. The entire manuscript was revised, and the English was improved.

Reviewer 2 Report
Comments and Suggestions for Authors
The paper entitled: „Continuous treatment of refractory wastewater from research and teaching laboratories via supercritical water oxidation – Experimental results and modeling” deals with an experimental study that proposes the application of the supercritical water oxidation process for the treatment of complex refractory wastewaters generated in research and teaching laboratories.
I believe that the manuscript requires extensive revision to improve the quality of presentation.
Language control is also needed. In addition, there are quite a few editorial, stylistic and punctuation errors in the paper.
It is an interesting work, but it needs many corrections:
1) Abstract
· Please do not write an abstract this way. An abstract should be able to stand alone and be independent of the paper. The research problem should be well described and entail a brief introduction, problem statement, method, results, conclusion and recommendation. Moreover, an abstract need to state more clearly aim of your work as well. I suggest that the Authors should rephrase it.
2) Introduction
· The introduction is not very strong. It is very poor in information. It lacks an overview of previous research. It would be useful to state what led to the design and selection of this segment of the research, and what was conducted in previous studies that led to it. At this point, it would also be extremely important to highlight the novelty of the research. So please explain what the novelty of this study is compared to previous studies. This is very important.
· In addition, the information contained in lines 96 - 106 should only and exclusively make clear the purpose of the research. The information there is too detailed for an introduction, it should be in the next section.
3) Methods and materials
· I would suggest adding a graphic description of the research methodology, this always helps to illustrate to the reader the experiments carried out.
· In section 2.1, I would suggest adding a table with detailed characteristics of the refractory wastewater samples analysed.
· On line 120 include the information "The experimental apparatus was operated in continuous mode" - for what period of time?
· Lines 148-149 - "The aqueous solution samples were prepared by mixing distilled water with refractory effluent based on the proportions established in the experimental design." - what specific proportions are you referring to?
· Please clarify and supplement the information in the manuscript with what is new in the present research procedure. Experimental articles must contain elements of novelty, this is very important.
· Line 186 - on what basis is it concluded that the conditions presented are optimal?
4) Result and Discussion
· The results of the study should be reworded a bit and descriptions should be simplified. In this form this section is uneasy to read.
· The paper should clearly state what is completely new in the results presented. The discussion should highlight the newly opened problems and the need to solve them. This is completely missing. Innovative character of this publication should be clearly specified and emphasized. In this case, it is felt that this study does not bring anything new.
· In addition:
ü The quality of the presented drawings is quite poor, it should be improved.
ü Do the test results in Table 5 present averaged values of contamination indicators?
ü Why is the wastewater generated at the Federal University of Goiás - UFG discussed separately in Section 3.4 and not in tandem with the wastewater from the Department of Chemical Engineering at the State University of Maringá?
ü Wherever optimal conditions are discussed, this statement should be expanded with specific information.
5) Conclusion
· Please do not write conclusion this way. The reader should get the real conclusions at the end of the study. This section is now written as a summary - but what are the conclusions of your research? I suggest bulleting out specific conclusions.

Author Response
Reviewer #2:
General comment: The paper entitled: “Continuous treatment of refractory wastewater from research and teaching laboratories via supercritical water oxidation – Experimental results and modeling” deals with an experimental study that proposes the application of the supercritical water oxidation process for the treatment of complex refractory wastewaters generated in research and teaching laboratories. I believe that the manuscript requires extensive revision to improve the quality of presentation. Language control is also needed. In addition, there are quite a few editorial, stylistic and punctuation errors in the paper. It is an interesting work, but it needs many corrections.
Answer: We appreciate the good evaluation of our work and thank the reviewer for your comments and suggestions, which have helped us to produce a better version of the manuscript.
Specific comments:
1) Abstract: Please do not write an abstract this way. An abstract should be able to stand alone and be independent of the paper. The research problem should be well described and entail a brief introduction, problem statement, method, results, conclusion, and recommendation. Moreover, an abstract need to state more clearly aim of your work as well. I suggest that the Authors should rephrase it.
Answer: The entire abstract section was revised and rephased taking in account the suggestion made by the esteemed reviewer as follows:
Abstract: Teaching and research laboratories generate wastes of various compositions and volumes, ranging from diluted aqueous solutions to concentrated ones, which, due to milder self-regulation waste management policies, are carelessly discarded, with little attention given to the consequences on the environment and human health. In this sense, the current study proposes the application of the supercritical water oxidation (SCWO) process for the treatment of complex refractory wastewater generated in research and teaching laboratories of universities. The SCWO, which uses water in conditions above its critical point (T > 647.1 K, P > 22.1 MPa), is regarded as an environmentally neutral process, uniquely adequate for the degradation of highly toxic and bio-refractory organic compounds. Initially, the wastewater samples were characterized via headspace gas chromatography coupled with mass spectrometry. Then, using a continuous tubular reactor, the selected operational parameters were optimized by a Taguchi L9 experimental design aiming to maximize the total organic carbon reduction. Initially, in a continuous system reactor the operational parameters were optimized by a Taguchi L9 experimental design aiming to maximize the total organic carbon reduction. Moreover, the SCWO treatment products, both liquid and gaseous, were carefully characterized. Under optimized conditions, that is, temperature of 823.15 K, feed flowrate of 10 mL min-1, oxidizing ratio of 1.5 (50% excess over the oxygen stoichiometric ratio) and sample concentration of 30%, a TOC, COD, and BOD reduction of 99.9%. 91.5% and 99.2% were achieved, respectively. During the treatment process, only CO2, methane and hydrogen were identified in the gaseous phase. Furthermore, the developed methodology was applied for the treatment of wastewater samples generated in another research laboratory and a TOC reduction of 99.5% was achieved, reinforcing the process robustness. A thermodynamic analysis of SCWO treatment of laboratory wastewater under isothermal conditions was performed using the Gibbs energy minimization methodology with the aid of the GAMS® 23.9.5. (General Algebraic Modeling System) software and the CONOPT 4 solver. Therefore, the results showed that SCWO could be efficiently applied for the treatment of wastewater generated by different teaching and research laboratories without the production of harmful gases and the addition of hazardous chemicals.
2) Introduction: The introduction is not very strong. It is very poor in information. It lacks an overview of previous research. It would be useful to state what led to the design and selection of this segment of the research, and what was conducted in previous studies that led to it. At this point, it would also be extremely important to highlight the novelty of the research. So please explain what the novelty of this study is compared to previous studies. This is very important. In addition, the information contained in lines 96 - 106 should only and exclusively make clear the purpose of the research. The information there is too detailed for an introduction, it should be in the next section.
Answer: An overview of previous research on the field was included as two additional paragraphs on the introduction section. Additionally, the information in lines 96 -106 were revised and rewritten as a simple statement of the purpose of the research. Finally, the novelty of the research was highlighted within the same statement. The suggestions were incorporated as follows:
The degradation of different types of dyes and other organic substances (imidazoline, acetic acid and dimethyl coco-benzyl ammonium-chloride) frequently found in the textile industry wastewater was investigated by Sogur and Askgun (2010) [20]. Using a tubular reactor, the system reached a complete TOC removal (100%) at a temperature of 823,15 K in the presence of H2O2 (17.73 mmol L−1) and a residence time of 10 s.
Youngprasert et al. (2010) investigated the degradation of model laboratory wastewater containing acetonitrile via supercritical water oxidation using a compact sized tubular reactor (internal volume of 4.71 mL) [21]. Manganese dioxide and H2O2 were used as the catalyst and oxidant, respectively. The complete oxidation of the acetonitrile was achieved at 673.15 K, 25 MPa and feed flow rate of 2 mL min-1. N2, CO2 and CO were observed as the main components of the gaseous phase. Ferreira-pinto et al. (2016) reported experimental data on TOC reduction via SCWO of a model dairy industry wastewater (lactose). Under constant pressure of 22.5 MPa, feed flow rate of 5 mg L-1, temperature of 823.15 K and presence of hydrogen peroxide, a continuous tubular achieved a TOC reduction of 99% [22].
Roshchin et al. (2017) observed that the supercritical water degradation of organic compounds, specifically well-known pesticides such as dichlorodiphenyltrichloroethane (DDT), hexachlorobenzene (HCB), and hexachlorocyclohexane (HCH), is highly dependent on the presence of an oxidizing agent [23]. The addition of air as an oxidizing agent enhanced the degree of decomposition of DDT, HCB and HCH from 75%, 52%, and 40% (at 923.15 K) to 99.8%, 99.7% and 99.9% (at 823.15 K), respectively. Mylapilli and Reddy (2019) studied the supercritical water oxidation of pharmaceutical industry wastewater containing analgesic, antibiotic, antipyretics, and antifungal with initial total organic carbon (TOC) concentration of 2,017 mg L-1. Under optimal conditions, that is, temperature of 823.15 K, pressure of 23 MPa, residence time of 60 s, and presence of H2O2, a reduction of 97.8% of the TOC was achieved [13].
In this sense, it remains clear that the recent research on the use of the supercritical gasification for treatment purposes have been focusing on wastewater generated by large scale industries [16]. On the other hand, when laboratory waste was investigated, only model solutions were assessed at a reduced scale and extremely low concentrations [21]. Therefore, the current study proposed the application of the SCWO for the continuous treatment of real organic refractory wastewater generated at research/teaching laboratories of two Brazilian Universities. Additionally, a thermodynamic simulation was conducted to determine the multiple component/phase system equilibrium.
3) Methods and materials: I would suggest adding a graphic description of the research methodology, this always helps to illustrate to the reader the experiments carried out.
Answer: A graphic description of the research methodology was added to the main manuscript as follows:
A graphic description of the research methodology is presented in Figure 2.
Figure 2. Graphic description of the research methodology.
4) In section 2.1, I would suggest adding a table with detailed characteristics of the refractory wastewater samples analysed.
Answer: A table with all detailed characteristics of the refractory wastewater samples analysed is available at the section 3.2. However, a summary of the main physico-chemical parameter of the untreated wastewater samples was added as suggested:
The detailed characterization of the refractory wastewater collected at the Department of Chemical Engineering at the State University of Maringá (Brazil) is summarized in Table 1.
Table 1. Main physico-chemical parameters of refractory wastewater samples.
Parameters
Concentration (mg L-1)
TOC
47980
COD
204876
BOD
489056
5) On line 120 include the information "The experimental apparatus was operated in continuous mode" - for what period of time?
Answer: The information was added as follows:
In general, after the desired operation conditions of feed flow rate, temperature, and pressure were reached, the system was operated over a period of 1 h for each experimental run to allow the accumulation of the liquid treated solution and analysis of the gaseous products.
6) Lines 148-149 - "The aqueous solution samples were prepared by mixing distilled water with refractory effluent based on the proportions established in the experimental design." - what specific proportions are you referring to?
Answer: The information was added as follows:
The aqueous solution samples were prepared by mixing distilled water with refractory effluent based on the proportions established in the experimental design, that is, 10, 20 or 30% by weight.
7) Please clarify and supplement the information in the manuscript with what is new in the present research procedure. Experimental articles must contain elements of novelty, this is very important.
Answer: As suggested earlier, the novelty was added in the introduction section as follows:
In this sense, remains clear that the recent research on the use of the supercritical gasification for treatment purposes have been focusing on wastewater generated by large scale industries [16]. On the other hand, when laboratory waste was investigated, only model solutions were assessed at a reduced scale and extremely low concentrations [21]. Therefore, the current study proposed the application of the SCWO for the continuous treatment of real organic refractory wastewater generated at research/teaching laboratories of two Brazilian Universities. Additionally, a thermodynamic simulation was conducted to determine the multiple component/phase system equilibrium.
8) Line 186 - on what basis is it concluded that the conditions presented are optimal?
Answer: According to Taguchi's optimization, the highest RTOC (selected response for maximization) could be achieved at a temperature of 823.15 K, feed rate of 10 mL min-1, oxidizing ratio of 1.5 and sample concentration of 30%. To validate the experimental design and confirm the robustness of the methodology employed, three additional experiments were conducted under the optimized conditions and an average RTOC of 99.9% ± 0.1 was achieved, confirming the Taguchi's method prediction.
9) Result and Discussion: The results of the study should be reworded a bit and descriptions should be simplified. In this form this section is uneasy to read. The paper should clearly state what is completely new in the results presented. The discussion should highlight the newly opened problems and the need to solve them. This is completely missing. Innovative character of this publication should be clearly specified and emphasized. In this case, it is felt that this study does not bring anything new.
Answer: The entire results and discussion section was revised; the text was reworded for simplicity and the descriptions were simplified. The novelty of the paper, although not stated in one simple sentence, is well mentioned across the text, for instance: i) the efficient treatment of highly concentrated (TOC ⁓ 48000) real wastewater samples generated at two different laboratories using a “pilot-scale” continuous tubular reactor; ii) Complete characterization of the wastewater samples as well as the gaseous and liquid products generated during the SCWO treatment process; iii) Presentation of the reaction mechanisms involving organic matter oxidation/degradation under supercritical water conditions and presence of oxidant substances; iv) Thermodynamic simulation of the SCWO treatment of laboratory wastewater and comparison with experimental results. These findings were added to the conclusion section (as bullet points) as suggested by the esteemed reviewer.
10) The quality of the presented drawings is quite poor; it should be improved.
Answer: All figures were revised, and their quality were improved.
11) Do the test results in Table 5 present averaged values of contamination indicators?
Answer: Yes, the reviewer is correct. Under optimal conditions (temperature of 823.15 K, feed rate of 10 mL min-1, oxidizing ratio of 1.5 and sample concentration of 30%), an experimental run was conducted in triplicate. The liquid products obtained in all three runs were mixed and then analyzed. Therefore, the results represent the mean value observed for the treated samples.
12) Why is the wastewater generated at the Federal University of Goiás - UFG discussed separately in Section 3.4 and not in tandem with the wastewater from the Department of Chemical Engineering at the State University of Maringá?
Answer: Usually, papers focusing on the application of SCWO for treatment purposes evaluate the wastewater generated in a single location and, sometimes, only model solutions (not real ones). In this sense, we decided to apply the SCWO for the treatment of real wastewater and asses the methodology robustness by processing the samples from different locations. To highlight this novelty, the results were discussed in a separate section.
13) Wherever optimal conditions are discussed, this statement should be expanded with specific information.
Answer: The specific optimal conditions information was added whenever optimal conditions were discussed.
14) Conclusion: Please do not write conclusion this way. The reader should get the real conclusions at the end of the study. This section is now written as a summary - but what are the conclusions of your research? I suggest bulleting out specific conclusions.
Answer: The conclusions of the study was added as follows:
In summary: 1. the efficient treatment via SCWO of highly concentrated (TOC ⁓ 48000) real wastewater samples was achieved; 2. The robustness of the methodology was proved by its successful application on wastewater samples generated at two different laboratories. 3. The formation of harmful gases, such as NO and NOx, were not observed; 4. The generation of ammoniacal nitrogen during the SCWO must be investigated; and 5. The thermodynamic simulation of the SCWO treatment of laboratory wastewater corroborated with the experimental results.

Reviewer 3 Report
Comments and Suggestions for Authors
Dear Authors,
the topic of disposal of chemical wastewater generated in laboratories is very important. However, minor changes are still required before final publication. See details below.
Were the studies on reduction of TOC (RTOC) in the treatment of refractory organic pollutants via supercritical water oxidation conducted once? Results should be reported as mean with standard deviation. with at least 3 repetitions.
Abstract – line 33 - Please explain the abbreviation ScWO.
Line 147 – temperature should be in unit “K”.
Equation 3 - Please write the multiplication sign, not “*”.
Line 442 - Correlation matrix is not figure but table. Where is C2H6 and C2H4 in the composition of the gas that the authors write about in lines 164-165.
In Figure 8b, the graph for methane, nitrogen and carbon monoxide is hardly visible.
Line 470 - How were reduction of COD and DBO calculated, which the authors write about in the conclusions?
Line 32 and 475- In table 4 is 99.4%.
Instead of “Eq. 1” ect. please write only (1).
Author Response
Reviewer #3:
General comment: Dear Authors, the topic of disposal of chemical wastewater generated in laboratories is very important. However, minor changes are still required before final publication. See details below.
Answer: We are grateful for the favorable evaluation of our work and the valuable suggestions for improving the paper.
Specific comments:
1) Were the studies on reduction of TOC (RTOC) in the treatment of refractory organic pollutants via supercritical water oxidation conducted once? Results should be reported as mean with standard deviation. with at least 3 repetitions.
Answer: Experiments were conducted in a continuous mode. Under the steady-state operation conditions of feed flow rate, temperature, and pressure, the treated liquid sample was accumulated over a period of 1 h. Thus, TOC measurements (n = 3) were performed in an aliquot representative for the test performed. The standard deviations of TOC measurements and uncertainties are now reported in the manuscript, as can be seen in Table 6.
Table 6. Physico-chemical parameters of the raw and treated wastewater samples under optimized SCWO conditions.
Parameters1
Sample
Reduction (%)
Raw
Treated2
Uncertainty3
TOC
47980 ± 116
60.1 ± 0.24
-
99.9
COD
204876
17458
0.060
91.5
BOD
489056
4162.5
0.145
99.2
Nitrite
0
0
0.030
-
Nitrate
0
0.2
0.004
-
N – NH3
12.3
362.8
-
-
Aluminum (Al)
0.03
0.01
0.0023
66.7
Calcium (Ca)
2.3
1.3
0.003
43.5
Chromium (Cr)
<
0.4
0.003
-
Iron (Fe)
0.02
<
0.003
-
Potassium (K)
7.4
0.6
0.004
91.9
Magnesium (Mg)
0.5
0.2
0.001
60
Sodium (Na)
11.5
2.75
0.005
76.1
Nickel (Ni)
0.004
0.05
0.004
-
Sulfur (S)
1152.8
21.5
0.0002
98.1
Zinc (Zn)
0.08
<
0.006
-
1 Treatment conditions: temperature (823.15 K), feed flow rate (10 mL min-1), and H2O2 ratio of 1.5.
2 Uncertainty = Expanded uncertainty (U), which is based on the combined standard uncertainty, with a 95% confidence level.
3 Regulated limit values are expressed in mg L-1.
2) Abstract – line 33 - Please explain the abbreviation SCWO.
Answer: The abbreviation definition has been added.
3) Line 147 – temperature should be in unit “K”.
Answer: The temperature was converted to Kelvin.
4) Equation 3 - Please write the multiplication sign, not “*”.
Answer: Thanks. It was revised.
5) Line 442 - Correlation matrix is not figure but table. Where is C2H6 and C2H4 in the composition of the gas that the authors write about in lines 164-165.
Answer: The nomenclature was revised. Regarding C2H6 and C2H4 compounds, they were present in the gaseous mix calibration standard used in the GC equipment. In other words, a quantitative determination could be applied for H2, CO, CO2, CH4, C2H6, and C2H4 gases, however, C2H6 and C2H4 were not found in the gaseous phase generated by the SCWO process. To make this clear, this section in materials and methods was rewritten, as can be seen bellow:
The composition of the gas generated during the degradation of the contaminants present in the target effluent was determined by gas chromatography (GC, ThermoFinnigan TRACE GC) equipped with a thermal conductivity and flame ionization detectors (TCD and FID), a 10-way valve system and a Porapak column in series with a 13X Molecular Sieve. The analyzes were conducted in isothermal mode with the columns at 328.15 K and the detector at 403.15 K. Argon was used as carrier gas at constant flow rate. A standard gas mixture with the composition (v/v) of H2 (50.01%), CO2 (2.04%), C2H4 (9.95%), C2H6 (10.02%), N2 (21.11%), CH4 (4.86%) and CO (2.01%) was used in the equipment calibration.
6) In Figure 8b, the graph for methane, nitrogen and carbon monoxide is hardly visible.
Answer: The graphs were reworked to facilitate data visualization, as depicted in Figure 8.
7) Line 470 - How were reduction of COD and DBO calculated, which the authors write about in the conclusions?
Answer: The parameters used in the calculation of the reduction of COD and BOD were added in the experimental section of the manuscript, as follows:
The reduction of COD was calculated according to Equation 4.
(4)
Where and are the initial and final concentration of COD, respectively, in the inlet and outlet of the SCWO reactor.
The reduction of BOD was calculated according to Equation 5.
(5)
Where and are the initial and final concentration of BOD, respectively, in the inlet and outlet of the SCWO reactor.
8) Line 32 and 475- In table 4 is 99.4%.
Answer: Three additional experiments were performed to validate the optimized conditions determined by Taguchi's methodology. In the optimal conditions, a RTOC of 99.9 % ± 0.1 was achieved, which corresponds to the value reported in the lines mentioned. The validation of the optimized conditions is reported in section 3.1, as follows:
In this study, according to Taguchi's optimization, the highest RTOC could be achieved at a temperature of 823.15 K, feed rate of 10 mL min-1, oxidizing ratio of 1.5 and sample concentration of 30%. To validate the experimental design and confirm the robustness of the methodology employed, three additional experiments were conducted under the optimized conditions and an average RTOC of 99.9% ± 0.1 was achieved.
9) Instead of “Eq. 1” ect. please write only (1).
Answer: It was revised.

Reviewer 4 Report
Comments and Suggestions for Authors
Article entitled Continuous treatment of refractory wastewater from research and teaching laboratories via supercritical water oxidation – Experimental results and modeling written by Mariana Bisinotto Pereira, Guilherme Botelho Meireles de Souza, Isabela Milhomem Dias, Julles Mitoura dos Santos Júnior, Antônio Carlos Daltro de Freitas, Jose M. Abelleira-Pereira, Christian Gonçalves Alonso, Lucio Cardozo-Filho and Reginaldo Guirardello and submitted to Water journal as a draft no water-2690439 deals with an important issue of wastewater treatment.
Article is in journal’s scope. Therefore, it could be considered for publication in Water journal. As English is not my native language, I am not able to assess language correctness. However, while reading, I found some statements missing, confusing or unclear. Below I enclose the list of my comments.
Literature review in introduction section is not deep enough. Current state of the knowledge according to SCWO was not provided.
Mechanism of SCWO was not presented.
Current state of the knowledge regarding wastewater from laboratories was not provided.
What was the aim of the research as well as the novelty? It should be clearly stated at the end of introduction.
How representative is the sample? How does the method of collecting it relate to the typical properties of this wastewater? Does this wastewater have a constant composition?
H2O2 is well known COD measurement disruptor. How the Authors solved this problem.
Composition of treated wastewater is not provided. Some info about organic substances is given. What about inorganic ones? Could they affect the treatment? Some info is in table 5 - far too late, but eg chlorides content is not given.
With GC-MS and HPLC it is only possible to determine compounds on trace concentrations. What about substances on high concentrations (in mg/L)?
What was the amount of each pollutant?
How do the Authors explain the removal of metals (how they were determined). Similarly, what is the mechanism for removing salinity?
Based on my comments and general impression I suggest major revision.
Author Response
Reviewer #4:
General comment: Article entitled Continuous treatment of refractory wastewater from research and teaching laboratories via supercritical water oxidation – Experimental results and modeling written by Mariana Bisinotto Pereira, Guilherme Botelho Meireles de Souza, Isabela Milhomem Dias, Julles Mitoura dos Santos Júnior, Antônio Carlos Daltro de Freitas, Jose M. Abelleira-Pereira, Christian Gonçalves Alonso, Lucio Cardozo-Filho and Reginaldo Guirardello and submitted to Water journal as a draft no water-2690439 deals with an important issue of wastewater treatment. Article is in journal’s scope. Therefore, it could be considered for publication in Water journal. As English is not my native language, I am not able to assess language correctness. However, while reading, I found some statements missing, confusing or unclear. Below I enclose the list of my comments.
Answer: We are grateful for your positive assessment of our paper and the valuable suggestions for improvement.
Specific comments:
1) Literature review in introduction section is not deep enough. Current state of the knowledge according to SCWO was not provided.
Answer: Thanks for pointing out this aspect. To address this point, we added an introduction of the state-of-the-art about the use of SCWO in the treatment of refractory and ubiquitous molecules in wastewater. More than one referees raised concerns about this point. The paragraphs added can be seen below:
The degradation of different types of dyes and other organic substances (imidazoline, acetic acid and dimethyl coco-benzyl ammonium-chloride) frequently found in the textile industry wastewater was investigated by Sogur and Askgun (2010) [20]. Using a tubular reactor, the system reached a complete TOC removal (100%) at a temperature of 823,15 K in the presence of H2O2 (17.73 mmol L−1) and a residence time of 10 s.
Youngprasert et al. (2010) investigated the degradation of model laboratory wastewater containing acetonitrile via supercritical water oxidation using a compact sized tubular reactor (internal volume of 4.71 mL) [21]. Manganese dioxide and H2O2 were used as the catalyst and oxidant, respectively. The complete oxidation of the acetonitrile was achieved at 673.15 K, 25 MPa and feed flow rate of 2 mL min-1. N2, CO2 and CO were observed as the main components of the gaseous phase. Ferreira-pinto et al. (2016) reported experimental data on TOC reduction via SCWO of a model dairy industry wastewater (lactose). Under constant pressure of 22.5 MPa, feed flow rate of 5 mg L-1, temperature of 823.15 K and presence of hydrogen peroxide, a continuous tubular achieved a TOC reduction of 99% [22].
Roshchin et al. (2017) observed that the supercritical water degradation of organic compounds, specifically well-known pesticides such as dichlorodiphenyltrichloroethane (DDT), hexachlorobenzene (HCB), and hexachlorocyclohexane (HCH), is highly dependent on the presence of an oxidizing agent [23]. The addition of air as an oxidizing agent enhanced the degree of decomposition of DDT, HCB and HCH from 75%, 52%, and 40% (at 923.15 K) to 99.8%, 99.7% and 99.9% (at 823.15 K), respectively. Mylapilli and Reddy (2019) studied the supercritical water oxidation of pharmaceutical industry wastewater containing analgesic, antibiotic, antipyretics, and antifungal with initial total organic carbon (TOC) concentration of 2,017 mg L-1. Under optimal conditions, that is, temperature of 823.15 K, pressure of 23 MPa, residence time of 60 s, and presence of H2O2, a reduction of 97.8% of the TOC was achieved [13].
In this sense, it remains clear that the recent research on the use of the supercritical gasification for treatment purposes have been focusing on wastewater generated by large scale industries [16]. On the other hand, when laboratory waste was investigated, only model solutions were assessed at a reduced scale and extremely low concentrations [21]. Therefore, the current study proposed the application of the SCWO for the continuous treatment of real organic refractory wastewater generated at research/teaching laboratories of two Brazilian Universities. Additionally, a thermodynamic simulation was conducted to determine the multiple component/phase system equilibrium.
2) Mechanism of SCWO was not presented.
Answer: Regarding this point, it was introduced an overall reaction mechanism involving organic matter oxidation/gasification under supercritical water conditions.
Regarding reaction mechanisms involving organic matter oxidation/degradation under supercritical water conditions, monitoring such reactions experimentally is difficult due to extreme supercritical reaction conditions. Literature-based mechanisms are promoted by radical species, which occur by initiation, propagation, and termination stages. Equations 11, 12, 13, and 14 shows some mechanism for radical species formation.
H2O → H • + OH •
(11)
H2O2 → 2OH •
(12)
2 H2O2 → 2 H2O + O2
(13)
H • + O2 → HO2 •
(14)
According to Li et al. (2020), when H2O2 is used as an oxidant, the hydrogen abstraction phase occurs according to the reactions presented in Equations 15, 16 and 17 [35].
OH• + H2O2 → H2O + HO2 •
(15)
OH • + HO2 • → H2O + O2
(16)
HO2 • + HO2 • → H2O2 + O2
(17)
Then, in the propagation phase, hydrogen, hydroxyl, and hydroperoxyl radicals decompose organic compounds into new radicals, as shown in Equations 18 – 21.
R • + O2 → RO2
(18)
RO2 • + RH → ROOH + R •
(19)
RO2 • → HOOR
(20)
R • → R • + C → RH
(21)
Finally, in the termination stage, free radicals interact to generate novel compounds, usually resulting in the formation of species characterized by simple molecular structures, as evidenced in Equations 22 – 25.
R • + R • → R – R
(22)
R • + RO • → ROR
(23)
RO • + RO • → ROOR
(24)
R • + ROO • → ROOR
(25)
The gas production during supercritical water processes is characterized by its inherent complexity, involving a succession of physical transformations and numerous chemical reactions occurring within the reactor, as noted by Hantoko et al. (2018) [36]. In summary, the comprehensive gasification process is represented by the overall reaction depicted in Equation 26.
CHxOy + (2–y)H2O → CO2 + (2–y + x/2)H2
(26)
3) Current state of the knowledge regarding wastewater from laboratories was not provided.
Answer: As requested by more than one reviewer, the state-of-the-art about the use of SCWO for the treatment of laboratory wastewater was added to the introduction section. Moreover, comparison with published studies that evaluated similar substances is present throughout the manuscript as follows:
Yang et al. (2018) evaluated the use of SCWO in the decomposition of 44 nitrogenous compounds and achieved efficiencies greater than 80%, in terms of TOC removal for all compounds evaluated at temperatures up to 823.15 K [38]. The researchers identified gaseous nitrogen, organic nitrogen, ammoniacal nitrogen and nitrate as the main nitrogen-containing compounds after treatment. In accordance with the results available in the literature, ammoniacal nitrogen (12.3 mg L-1 → 362.8 mg L-1) was observed as the main by-product of the treatment of organic nitrogenous compounds via oxidation in supercritical water. According to Bermejo et al. (2008) temperatures above 973.15 K (higher than the Inconel 625 rating at the typical working pressures for SCWO) are necessary for the degradation of ammonia via oxidation in supercritical water, reaching up to 1073.15 K for an ammonia concentration of 7% wt [39]. Additionally, another strategy that could be used for the enhancement of the ammonium removal during the SCWO process, is the use of organic solvents, such as isopropanol, as co-fuels [40].
According to Chakinala et al. (2013), methanol is considered a stable compound at temperatures below 873.15 K in the absence of a catalyst [46]. The decomposition of methanol can occur through different routes. One route involves the abstraction of a hydrogen atom from an oxygen atom, resulting in the generation of a H3CO • radical. This radical can decompose into formaldehyde and release an H • radical. Then, formaldehyde decomposes directly into CO and H2, or, intermediately, it can oxidize into formic acid, which decomposes into CO2 and H2. Another possible initiation route is the hydrogen abstraction on the α-carbon atom, which can also lead to the formation of formaldehyde intermediates. Cleavage of the C–O bond is a dehydration pathway that occurs when an H • radical reacts with an OH group present in the compound, forming water and a CH3 • radical, which can then lead to the formation of methane.
4) What was the aim of the research as well as the novelty? It should be clearly stated at the end of introduction.
Answer: Improvements were made to the manuscript and a new sentence was added addressing this point.
In this sense, it remains clear that the recent research on the use of the supercritical gasification for treatment purposes have been focusing on wastewater generated by large scale industries [16]. On the other hand, when laboratory waste was investigated, only model solutions were assessed at a reduced scale and extremely low concentrations [21]. Therefore, the current study proposed the application of the SCWO for the continuous treatment of real organic refractory wastewater generated at research/teaching laboratories of two Brazilian Universities. Additionally, a thermodynamic simulation was conducted to determine the multiple component/phase system equilibrium.
5) How representative is the sample? How does the method of collecting it relate to the typical properties of this wastewater? Does this wastewater have a constant composition?
Answer: Samples were collected after an accumulative residue from HPLC analysis for a period higher than six months. However, due to the long period of wastewater accumulation, several analytes can be present in the samples collected, which represents a large variability of compounds at trace level. The samples assessed wer representative of this whole accumulation period. To evaluate the variability of the sample composition, among majority compounds, samples also were collected in laboratories of two distinct universities, where the lab routine are different. As expected, the majority of compound determined were similar, in other words, they were representative for the commonly methodologies used in HPLC, which methanol, acetonitrile and chloroform solvent are used.
6) H2O2 is well known COD measurement disruptor. How the Authors solved this problem.
Answer: Great note. In our case, the presence of H2O2 was not a problem in the COD measurement for two main reasons. First, because the raw wastewater samples were analyzed before of H2O2 addition. Second, samples containing the oxidizing agent were analyzed after treatment, which means, they were processed under supercritical conditions, and had H2O2 fully reduced or degraded due to high-temperature exposure in the SCWO process.
7) Composition of treated wastewater is not provided. Some info about organic substances is given. What about inorganic ones? Could they affect the treatment? Some info is in table 5 - far too late, but eg chlorides content is not given.
Answer: Thanks for raising this question. Improvements were made in the manuscript discussion and the effects of inorganic compounds also were added and shortly discussed. Our first data presentation corresponds to the RTOC, which was used in the operational parameter optimization. Then, considering the optimized conditions of the reaction, a detailed physicochemical characterization for both untreated and treated samples was provided in Table 6. However, unfortunately, measurements of chloride content were not performed. Some of improvements are highlighted below:
This is possible due to the distinct attributes exhibited by water under supercritical conditions. Within this context, there is a notable reduction in the dielectric constant (ɛ) and ionic product (Kw) values, consequently leading to a considerable diminution in the solubility of inorganic substances, including oxides and salts, which may explain the removal of metals observed after SCWO treatment [37].
8) With GC-MS and HPLC it is only possible to determine compounds on trace concentrations. What about substances on high concentrations (in mg/L)?
Answer: GC-MS is a highly sensitive analytical method employed for the determination of diverse compounds. In this study, the wastewater exhibited a complex composition. This complexity is due to wastewater origin, which is from a university laboratory with multifaceted applications and the accumulation of samples over an extended period. The laboratory's routine activities provided clear indications that the main contributors to the elevated Total Organic Carbon (TOC) concentration, the key parameter for process optimization, were organic solvents. This observation was substantiated through effluent characterization, which revealed methanol, acetonitrile, and chloroform as the predominant constituents.
9) What was the amount of each pollutant?
Answer: Quantitative analyses were primarily focused on concentration parameters related to both organic and inorganic constituents, including BOD, COD, TOC, and various physicochemical parameters. Additional quantitative data can be found in Table 6. The degradation of compounds, specifically organic solvents, was assessed through qualitative analysis, which determined the presence or absence of these compounds. Nevertheless, the notable reduction in TOC concentration strongly supports the findings of the qualitative analysis, suggesting that the most of organic solvents were degraded.
10) How do the Authors explain the removal of metals (how they were determined). Similarly, what is the mechanism for removing salinity?
Answer: Metals determination was performed in the liquid samples before and after treatment. Regarding the metals removal we believe that some of metals were retained inside the reactor. This hypothesis is supported by the distinct attributes exhibited by water under supercritical conditions. Within this context, there is a notable reduction in the dielectric constant (ɛ) and ionic product (Kw) values, consequently leading to a considerable diminution in the solubility of inorganic substances, including oxides and salts, which explain the removal of metals observed after SCWO treatment.
This is possible due to the distinct attributes exhibited by water under supercritical conditions. Within this context, there is a notable reduction in the dielectric constant (ɛ) and ionic product (Kw) values, consequently leading to a considerable diminution in the solubility of inorganic substances, including oxides and salts, which may explain the removal of metals observed after SCWO treatment [37].
11) Based on my comments and general impression I suggest major revision.
Answer: The manuscript was fully revised as suggested.

Round 2
Reviewer 2 Report
Comments and Suggestions for Authors
The authors have corrected the critical remarks suggested in the review.
The paper is now in better condition than before - the corrections make difference. I recommend the paper for publication in the Water journal.
Reviewer 4 Report
Comments and Suggestions for Authors
This is my second review of this manuscript. The Authors answered all of my questions. Suggested corrections have been applied. Second version is far better than the first one. I suggest to accept this manuscript in its current form.